



# 1 Effect of elevated $p$CO$_2$ on trace gas production during an

# 2 ocean acidification mesocosm experiment

Sheng-Hui Zhang[1,2], Qiong-Yao Ding[1], Gui-Peng Yang[1*], Kun-Shan Gao[3], Hong-Hai Zhang[1],
Da-Wei Pan[2]
[1] Key Laboratory of Marine Chemistry Theory and Technology, Ministry of Education, Ocean University of China,
Qingdao 266100, China;
[2] Key Laboratory of Coastal Environmental Processes and Ecological Remediation, Yantai Institute of Coastal
Zone Research (YIC), Chinese Academy of Sciences(CAS); Shandong Provincial Key Laboratory of Coastal
Environmental Processes, YICCAS, Yantai Shandong 264003, P. R. China
[3] State Key Laboratory of Marine Environmental Science, Xiamen University, Xiamen, 361102, China
* Corresponding author:
Prof. Gui-Peng Yang
Key Laboratory of Marine Chemistry Theory and Technology
Ocean University of China
Qingdao 266100
China
E-mail: gpyang@mail.ouc.edu.cn
Tel: +86-532-66782657
Fax: +86-532-66782657





**Abstract**
A mesocosm experiment was conducted in Wuyuan Bay (Xiamen), China to investigate the effects of elevated
$p\mathrm{CO_2}$ on phytoplankton species and production of dimethylsulfide (DMS) and dimethylsulfoniopropionate (DMSP)
as well as four halocarbon compounds ($\mathrm{CHBrCl_2}$, $\mathrm{CH_3Br}$, $\mathrm{CH_2Br_2}$, and $\mathrm{CH_3I}$). Over a period of 5 weeks, *P.*
*tricornutum* outcompeted *T. weissflogii* and *E. huxleyi*, comprising more than 99% of the final biomass. During the
logarithmic growth phase (phase I), DMS concentrations in high $p\mathrm{CO_2}$ mesocosms (1000 μatm) were 28.2% lower
than those in low $p\mathrm{CO_2}$ mesocosms (400 μatm). Elevated $p\mathrm{CO_2}$ led to a delay in DMSP-consuming bacteria
attached to *T. weissflogii* and *P. tricornutum* and finally resulted in the delay of DMS concentration in the HC
treatment. Unlike DMS, the elevated $p\mathrm{CO_2}$ did not affect DMSP production ability of *T. weissflogii* or *P.*
*tricornutum* throughout the 5 week culture. A positive relationship was detected between $\mathrm{CH_3I}$ and *T. weissflogii*
and *P. tricornutum* during the experiment, and there was a 40.2% reduction in mean $\mathrm{CH_3I}$ concentrations in the
HC mesocosms. $\mathrm{CHBrCl_2}$, $\mathrm{CH_3Br}$, and $\mathrm{CH_2Br_2}$ concentrations did not increase with elevated chlorophyll *a* (Chl *a*)
concentrations compared with DMS(P) and $\mathrm{CH_3I}$, and there were no major peak in the HC or LC mesocosms. In
addition, no effect of elevated $p\mathrm{CO_2}$ was identified for any of the three bromocarbons.
**Keywords:** ocean acidification, dimethylsulfide (DMS), dimethylsulfoniopropionate (DMSP), halocarbon,
phytoplankton, bacteria







## 1. Introduction


As a result of human activity, anthropogenic emissions has increased the fugacity of atmospheric
carbon dioxide ($p$CO$_2$) from the pre-industrial value of 280 μatm to the present-day value of over
400 μatm, and these values will further increase to 800–1000 μatm by the end of this century
according to the Intergovernmental Panel on Climate Change (IPCC, 2014). The dissolution of
this excess CO$_2$ into the surface of the ocean directly affects the carbonate system and has lowered
the pH by 0.1 units, from 8.21 to 8.10 over the last 250 years. Further decreases of 0.3–0.4 pH
units are predicted by the end of this century (Doneyet al., 2009; Orr et al., 2005), which is
commonly referred to as ocean acidification (OA). The physiological and ecological aspects of the
phytoplankton response to this changing environment can potentially alter marine phytoplankton
community composition, community biomass, and feedback to biogeochemical cycles (Boyd and
Doney, 2002). These changes simultaneously have an impact on some volatile organic compounds
produced by marine phytoplankton (Liss et al., 2014; Liu et al., 2017), including the climatically
important trace gas dimethylsulfide (DMS) and a number of volatile halocarbon compounds.
DMS is the most important volatile sulfur compound produced from the algal secondary
metabolite dimethylsulfoniopropionate (DMSP) through complex biological interactions in marine
ecosystems (Stefels et al., 2007). Although it remains controversial, DMS and its by-products,
such as methanesulfonic acid and non-sea-salt sulfate, are suspected to have a prominent part in
climate feedback (Charlson et al., 1987; Quinn and Bates, 2011). The conversion of DMSP to
DMS is facilitated by several enzymes, including DMSP-lyase and acyl CoA transferase
(Kirkwood et al., 2010; Todd et al., 2007); these enzymes are mainly found in phytoplankton,
macroalgae, *Symbiodinium*, bacteria and fungi (de Souza and Yoch, 1995; Stefels and Dijkhuizen,





1996; Steinke and Kirst, 1996; Bacic and Yoch, 1998; Yost and Mitchelmore, 2009). Several
studies have already reported the sensitivity of DMS-production capability to ocean acidification.
Majority of these experimental studies revealed negative impact of decreasing pH on
DMS-production capability (Hopkins et al., 2010; Avgoustidi et al., 2012; Archer et al., 2013;
Webb et al.,2016), while others found either no effect or a positive effect (Vogt et al., 2008;
Hopkins and Archer, 2014). Several assumptions have been presented to explain these contrasting
results and attribute the pH-induced variation in DMS-production capability to altered physiology
of the algae cells or of bacterial DMSP degradation (Vogt et al., 2008; Hopkins et al., 2010,
Avgoustidi et al., 2012; Archer et al., 2013; Hopkins and Archer, 2014; Webb et al., 2015, 2016).
Halocarbons also play a significant role in the global climate because they are linked to
tropospheric and stratospheric ozone depletion and a synergistic effect of chlorine and bromine
species has been reported that they may account for approximately 20% of the polar stratospheric
ozone depletion (Roy et al., 2011). In addition, iodocarbons can release atomic iodine (I) quickly
through photolysis in the atmospheric boundary layer and I atoms are very efficient in the catalytic
removal of $O_3,$ which governs the lifetime of many climate relevant gases including methane ($CH_4$)
and DMS (Jenkins et al., 1991). Compared with DMS, limited attention was received about the
effect of OA on halocarbon concentrations. Hopkins et al. (2010) and Webb (2015) measured
lower concentrations of several iodocarbons, while bromocarbons were unaffected by elevated
$p$CO$_2$ through two acidification experiments. In addition, an additional mesocosm study did not
elicit significant differences from any halocarbon compounds at up to 1,400 µatm $p$CO$_2$ (Hopkins
et al., 2013).

The combined picture arising from existing studies is that the response of communities to OA



is not predictable and requires further study. Here, we report a mesocosm experiment conducted to
study the influence of elevated $p$CO$_2$ on the biogeochemical cycle of a laboratory-cultured
artificial phytoplankton community of diatoms and coccolithophores that had been previously
examined for the response to elevated $p$CO$_2$. Our objective was to assess how changes in the
phytoplankton community driven by changes in $p$CO$_2$ impact dimethyl sulfur compounds and
halocarbons (including CH$_3$I, CHBrCl$_2$, CH$_3$Br, and CH$_2$Br$_2$) release.
**2. Experimental method**
*2.1 General experimental device*
The mesocosm experiments were carried out on a floating platform at the Facility for Ocean
Acidification Impacts Study of Xiamen University (FOANIC-XMU, 24.52 °N, 117.18 °E) in Wu
Yuan Bay, Xiamen (for full technical details of the mesocosms, see Liu et al. 2017). Six
cylindrical transparent thermoplastic polyurethane bags with domes were deployed along the
south side of the platform. The width and depth of each mesocosm bag was 1.5 m and 3 m,
respectively. Filtered (0.01 μm, achieved using an ultrafiltration water purifier, MU801-4T, Midea,
Guangdong, China) *in situ* seawater was pumped into the six bags simultaneously within 24 h. A
known amount of NaCl solution was added to each bag to calculate the exact volume of seawater
in the bags, according to a comparison of the salinity before and after adding salt (Czerny et al.,
2013). The initial *in situ* $p$CO$_2$ was about 650 μatm. To set the low and high $p$CO$_2$ levels, we
added Na$_2$CO$_3$ solution and CO$_2$ saturated seawater to the mesocosm bags to alter total alkalinity
and dissolved inorganic carbon (Gattuso et al., 2010; Riebesell et al., 2013). Subsequently, during
the whole experimental process, air at the ambient (400 μatm) and elevated $p$CO$_2$ (1000 μatm)
concentrations was continuously bubbled into the mesocosm bags using a CO$_2$ Enricher (CE-100B,





Wuhan Ruihua Instrument & Equipment Ltd., Wuhan, China). Because the seawater in the
mesocosm was filtered, the algae in the coastal environment and their attached bacteria were
removed and the trace gases produced in the environment did not influence the mesocosm trace
gas concentrations after the bags were sealed.
*2.2 Algal strains*
Three phytoplankton strains were inoculated into the mesocosm bags, at $4 \times 10^4$ cells $L^{-1}$ each *P.*
*tricornutum* (CCMA 106) and *T. weissflogii* (CCMA 102) were obtained from the Center for
Collections of Marine Bacteria and Phytoplankton of the State Key Laboratory of Marine
Environmental Science (Xiamen University). *P. tricornutum* was originally isolated from the
South China Sea in 2004 and *T. weissflogii* was isolated from Daya Bay in the coastal South China
Sea. *E. huxleyi* PML B92/11 was originally isolated in 1992 from the field station of the
University of Bergen (Raunefjorden; 60°18'N, 05°15'E).
*2.3 Sampling for DMS(P) and halocarbons*
DMS(P) and halocarbons samples were generally obtained from six mesocosms at 9 a.m., then all
collected samples were transported into a dark cool box back to the laboratory onshore for analyse
within 1 h. For DMS analysis, 2 mL sample was gently filtered through a 25 mm GF/F (glass fiber)
filter and transferred to a purge and trap system linked to a Shimadzu GC-2014 gas
chromatograph (Tokyo, Japan) equipped with a glass column packed with 10% DEGS on
Chromosorb W-AW-DMCS (3 m × 3 mm) and a flame photometric detector (FPD) (Zhang et al.,
2014). For total DMSP analysis, 10 mL water sample was fixed using 50 μL of 50 % $H_2SO_4$ and
sealed (Kiene and Slezak, 2006). After > 1 d preservation, DMSP samples were hydrolysed for 24
h with a pellet of KOH    (final pH > 13) to fully convert DMSP to DMS. Then, 2 mL hydrolysed





sample was carefully transferred to the purge and trap system mentioned above for extraction of
DMS. For halocarbons, 100 mL sample was purged at 40 ℃ with pure nitrogen at a flow rate of
100 mL min$^{-1}$ for 12 min using another purge and trap system coupled to an Agilent 6890 gas
chromatograph (Agilent Technologies, Palo Alto, CA, USA) equipped with an electron capture
detector (ECD) as well as a 60 m DB-624 capillary column (0.53 mm ID; film thickness, 3 μm)
(Yang et al., 2010). The analytical precision for duplicate measurements of DMS(P) and
halocarbons was > 10%.
*2.4 Measurements of chlorophyll a*
Chlorophyll *a* (Chl *a*) was measured in water samples (200–1,000 mL) collected every 2 d at 9
a.m. by filtering onto Whatman GF/F filters (25 mm). The filters were placed in 5 ml 100%
methanol overnight at 4 ℃ and centrifuged at 5000 r min$^{-1}$ for 10 min. The absorbance of the
supernatant (2.5 mL) was measured from 250 to 800 nm using a scanning spectrophotometer (DU
800, Beckman Coulter Inc., Brea, CA, USA). Chl *a* concentration was calculated according to the
equation reported by Porra (2002).
*2.5 Statistical analysis*
One-way analysis of variance (ANOVA), Tukey's test, and the two-sample *t*-test were carried out
to demonstrate the differences between treatments. A *p*-value < 0.05 was considered significant.
Relationships between DMS(P), halocarbons and a range of other parameters were detected using
Pearson's correlation analysis via SPSS 22.0 for Windows (SPSS Inc., Chicago, IL, USA).
**3. Results and Discussion**
*3.1 Temporal changes in pH, Chl a, P. tricornutum, T. weissflogii, and E. huxleyi during the*
*experiment*



During the experiment, the seawater in each mesocosm was well combined, and the temperature
and salinity were well controlled, with a mean of 16 $^{\circ}$C and 29 in all mesocosms, respectively
(Huang et al., 2018). Meanwhile, we observed significant differences in $p$CO$_2$ levels between the
two CO$_2$ treatments on days 0–11, but the differences disappeared with subsequent phytoplankton
growth (Fig. 1-A). The phytoplankton growth process was divided into three phases in terms of
variations in Chl $a$ concentrations (Fig. 1-B) in the mesocosm experiments: i) the logarithmic
growth phase (phase I, days 0–12), ii) a plateau phase (phase II, days 12–22, bloom period), and iii)
a secondary plateau phase (phase III, days 22–33) attained after a decline in biomass from a
maximum in phase II. The initial chemical parameters of the mesocosm experiment are shown in
Table 1. The initial mean dissolved nitrate (including NO$_3^-$ and NO$_2^-$), NH$_4^+$, PO$_4^{3-}$ and silicate
(SiO$_3^{2-}$) concentrations were 54 µmol L$^{-1}$, 20 µmol L$^{-1}$, 2.6 µmol L$^{-1}$ and 41 µmol L$^{-1}$ for the LC
treatment and 52 µmol L$^{-1}$, 21 µmol L$^{-1}$, 2.4 µmol L$^{-1}$ and 38 µmol L$^{-1}$ for the HC treatment,
respectively. The nutrient concentrations (NO$_3^-$, NO$_2^-$, NH$_4^+$ and phosphate) during phase I were
consumped rapidly and there concentrations were below or close to the detection limit during
phase II (Table 1). Meanwhile, Chl $a$ concentration increased rapidly and reached 109.9 and 108.6
mg L$^{-1}$ in the LC and HC treatments, respectively. In addition, although DIN (NH$_4^+$, NO$_3^-$, and
NO$_2^-$) and phosphate were depleted, Chl $a$ concentration in both treatments (biomass dominated
by $P.$ $tricornutum$) remained constant over days 12–22, and then declined over subsequent days as
shown in Liu et al. 2017.
$E.$ $huxleyi$ was only found in phase I and its maximal concentration reached 310 cells mL$^{-1}$
according to the results of microscopic inspection (Fig. 2-C). $T.$ $weissflogii$ was found throughout
the entire period in each bag, but the maximum concentration was 8,120 cells mL$^{-1}$, which was far





less than the concentration of *P. tricornutum* with a maximum cell density of about 1.5 million
cells mL$^{-1}$ (Fig. 2-A and Fig. 2-B). *P. tricornutum* accounted for at least 99% of all of the biomass
by the time the populations had entered the plateau phase (phase II). We did not detect any
significant enhancement in elevated $p$CO$_2$ due to the large variation. However, significant
differences between the two $p$CO$_2$ treatments were found on days 23 ($p = 0.006$) and 25 ($p = 0.007$)
(Fig. 2-A), when the cell concentration declined. Although we did not observe any difference
between the two $p$CO$_2$ treatments during the rapid growth period (days 8–15), a longer period of
persistent cell growth and a slower pace during the decrease in population size in phase II were
recorded under the HC condition compared to the LC condition (Fig. 2-A).
*3.2 Impact of* elevated *$p$CO$_2$ on DMS and DMSP production*
Several studies have already reported the sensitivity of DMS-production capability to decreases in
seawater pH. However, these studies did not come to a unified conclusion (Vogt et al., 2008;
Hopkins et al., 2010; Avgoustidi et al., 2012; Archer et al., 2013; Hopkins and Archer, 2014; Webb
et al.,2016). Fig. 3 (A-B) shows the mean DMS and DMSP concentrations for the HC and LC
treatments during the mesocosm experiment. At the beginning of the experiment, the mean DMS
and DMSP concentrations were low in both treatments due to the low concentrations of DMS and
DMSP in the original fjord water and possible loss during the filtration procedure. DMS and
DMSP showed slightly different trends during growth in the mesocosm experiment. The DMSP
concentrations in the HC and LC treatments increased significantly along with the increase in Chl
*a* and cell concentrations, and stayed relatively constant over the following days. A significant
positive relationship was observed between DMSP and phytoplankton in the experiment ($R^2 =$
0.92 $p < 0.01$ for *P. tricornutum*, $R^2 = 0.36$ $p < 0.01$ for *T. weissflogii* in LC treatment; $R^2 = 0.94$ $p$



< 0.01 for *P. tricornutum*, $R^2 = 0.36$ $p < 0.01$ for *T. weissflogii* in HC treatment). Mean
concentrations of DMS in the HC and LC treatments did not increase significantly (1.03 and 0.74
nmol $L^{-1}$ for the LC and HC treatments, respectively) during phase I, but began to increase rapidly
beginning on day 15. The two treatments peaked on days 25 (112.1 nmol $L^{-1}$) and 30 (101.9 nmol
$L^{-1}$), respectively, and then began to decrease during phase III. A significant positive relationship
was observed between DMS and phytoplankton throughout the experiment ($R^2 = 0.65$ $p < 0.01$ for
*P. tricornutum*, $R^2 = 0.80$ $p < 0.01$ for *T. weissflogii* in LC treatment; $R^2 = 0.54$ $p < 0.01$ for *P.*
*tricornutum*, $R^2 = 0.73$ $p < 0.01$ for *T. weissflogii* in HC treatment).

A significant 28.2% reduction in DMS concentration was detected in the HC treatment

compared with the LC treatment ($p = 0.016$) during phase I and this reduction in DMS
concentrations may be attributed to greater consumption of DMS and conversion to DMSO (Webb
et al., 2015). In contrast, no difference in mean DMSP concentrations was observed between the
two treatments, indicating that elevated $p$CO$_2$ had no significant influence on DMSP production in
*P. tricornutum* and *T. weissflogii* during this study. In addition, the peak DMS concentration in the
HC treatment was delayed 5 days relative to that in the LC treatment during phase II (Fig. 3-A).
This result has been observed in previous mesocosm experiments and it was attributed to small
scale shifts in community composition and succession that could not be identified with only a
once-daily measurement regime (Vogt et al., 2008; Webb et al., 2016). However, this phenomenon
can be explained in another straightforward way during this study. Previous studies have showed
that marine bacteria play a key role in DMS production and the efficiency of bacteria converting
DMSP to DMS may vary from 2 to 100% depending on the nutrient status of the bacteria and the
quantity of dissolved organic matter (Simó et al., 2002, 2009; Kiene et al., 1999, 2000). All of these





observations point to the importance of bacteria in DMS and DMSP dynamics. During the present
mesocosm experiment, DMSP concentrations in the LC treatment decreased slightly on day 23,
while the slight decrease appeared on day 29 in the HC treatment (Fig. 3-B). In addition, the time
that the DMSP concentration began to decrease was very close to the time when the highest DMS
concentration occurred in both treatments. Moreover, DMSP-consuming bacterial abundance
peaked on days 19 and 21 in the LC and HC treatments, respectively, as shown in Fig. S1 (Yu et
al., unpublished data). DMSP-consuming bacterial abundance was also delayed in the HC
mesocosm compared to that in the LC mesocosm. Taken together, we inferred that the elevated
$p$CO$_2$ first delayed growth of DMSP-consuming bacteria in the mesocosm, then the delayed
DMSP-consuming bacteria abundance postponed the DMSP degradation process, and eventually
delayed the DMS concentration in the HC treatment. In addition, considering that the algae and
their attached bacteria were removed through a filtering process before the experiment and the
unattached bacteria were maintained in a relatively constant concentration during this mesocosm
experiment (Huang et al., 2018), we further concluded that the elevated $p$CO$_2$ controlled DMS
concentrations mainly by affecting DMSP-consuming bacteria attached to *T. weissflogii* and *P.*
*tricornutum*. Moreover, the inhibition of elevated $p$CO$_2$ to DMSP-consuming bacteria might be
another important reason for the reduction of DMS in the HC treatment during phase I.
*3.3 Impact of elevated p*CO$_2$ *on halocarbon compounds*
The temporal development in CHBrCl$_2$, CH$_3$Br, and CH$_2$Br$_2$ concentrations is shown in Fig. 3
(C–E) and the temporal changes in their concentrations were substantially different from those of
DMS, DMSP, *T. weissflogii*, and *P. tricornutum*. The mean concentrations of CHBrCl$_2$, CH$_3$Br and
CH$_2$Br$_2$ for the entire experiment were 8.58, 7.85, and 5.13 pmol L$^{-1}$ in the LC treatment and 8.81,





9.73, and 6.27 pmol L$^{-1}$ in the HC treatment. The concentrations of $CHBrCl_2$, $CH_3Br$, and $CH_2Br_2$
did not increase with the Chl $a$ concentration compared with those of DMS and DMSP, and no
major peaks were detected in the mesocosms. In addition, no effect of elevated $p$CO$_2$ was
identified for any of the three bromocarbons, which compared well with previous mesocosm
findings (Hopkins et al., 2010, 2013; Webb, 2016). No clear correlation was observed between the
three bromocarbons and any of the measured algal groups, indicating that *T. weissflogii* and *P.*
*tricornutum* did not primarily release these three bromocarbons during the mesocosm experiment.
Previous studies have reported that large-size cyanobacteria, such as *Aphanizomenon flos*-aquae,
produce bromocarbons (Karlsson et al. 2008) and significant correlations between cyanobacterium
abundance and several bromocarbons have been reported in the Arabian Sea (Roy et al., 2011).
However, the filtration procedure led to the loss of cyanobacterium in the mesocosms and finally
resulted in low bromocarbon concentrations during the experiment, although *T. weissflogii* and *P.*
*tricornutum* abundances were high.

CH$_3$I prodution is usually involve to "biogenic", as it is released directly by macroalgae and

phytoplankton, and indirectly generated via a photochemical degradation with organic matter
(Moore and Zafiriou, 1994; Archer et al., 2007; Laturnus, 1995). The CH$_3$I concentrations in the
HC and LC treatments are shown in Fig. 3-F. The maximum CH$_3$I concentrations in the HC and
LC treatments were both observed on day 23 (12.61 and 8.78 pmol L$^{-1}$ for the LC and HC
treatments, respectively). A positive relationship was detected between CH$_3$I and Chl $a$ in both *T.*
*weissflogii* and *P. tricornutum* ($R^2 = 0.35$ $p < 0.01$ in LC treatment; $R^2 = 0.76$ $p < 0.01$ in HC
treatment for *P. tricornutum*; $R^2 = 0.48$ $p < 0.01$ in LC treatment; $R^2 = 0.48$ $p < 0.01$ in HC
treatment for *T. weissflogii*; $R^2 = 0.54$ $p < 0.01$ in LC treatment; $R^2 = 0.53$ $p < 0.01$ in HC



treatment for Chl *a*). This result agrees with previous mesocosm (Hopkins et al., 2013) and
laboratory experiments (Hughes et al., 2013; Manley and De La Cuesta, 1997) identifying diatoms
as significant producers of $CH_3I$. Morevover, similar to DMS, the maximum $CH_3I$ concentration
also occurred after the maxima of *T. weissflogii* and *P. tricornutum*, at about 4 d (Fig. 3-F). This
was similar to iodocarbon gases measured in a Norway mesocosm conducted by Hopkins et al.
(2010) and chloroiodomethane ($CH_2ClI$) concentrations measured in another Norway mesocosm
conducted by Wingenter et al. (2007). Furthermore, the $CH_3I$ concentrations measured in the HC
treatment were significantly lower than those measured in the LC treatment during the mesocosm,
which is in accord with the discoveries of Hopkins et al. (2010) and Webb et al. (2015) but in
contrast to the findings of Hopkins et al. (2013) and Webb et al. (2016). Throughout the mesocosm
experiment, there was a 40.2% reduction in the HC mesocosm compared to the LC mesocosm.
Considering that the phytoplankton species did not show significant differences in the HC and LC
treatments during the experiment, this reduction in the HC treatment was likely not caused by
phytoplankton. Apart from direct biological production via methyl transferase enzyme activity by
both phytoplankton and bacteria (Amachi et al., 2001), $CH_3I$ is produced from the breakdown of
higher molecular weight iodine-containing organic matter (Fenical, 1982) through photochemical
reactions between organic matter and light (Richter and Wallace, 2004). Both bacterial methyl
transferase enzyme activity and a photochemical reaction may have reduced the $CH_3I$
concentrations in the HC treatment but further experiments are needed to verify this result.
**4. Conclusions**
In this study, the effects of increased levels of $p\mathrm{CO_2}$ on marine DMS(P) and halocarbons release
were studied in a controlled mesocosm facility. A 28.2% reduction during the logarithmic growth




phase and a 5 d delay in DMS concentration was observed in the HC treatment due to the effect of
elevated $pCO_2$. Because the seawater in the mesocosm was filtered, the algae in the coastal
environment and their attached bacteria were removed and the trace gases produced in the
environment did not influence the mesocosm trace gas concentrations after the bags were sealed.
Therefore, we attribute this phenomenon to the DMSP-consuming bacteria attached to *P.*
*tricornutum* and *T. weissflogii*. More attention should be paid to the DMSP-consuming bacteria
attached to algae under different pH values in future studies. Three bromocarbons compounds
were not correlated with a range of biological parameters, as they were affected by the filtration
procedure and elevated $pCO_2$ had not effect on any of the three bromocarbons. The temporal
dynamics of $CH_3I$, combined with strong correlations with biological parameters, indicated
biological control of the concentrations of this gas. In addition, the production of $CH_3I$ was
sensitive to $pCO_2$, with a significant increase in $CH_3I$ concentration at higher $pCO_2$. However,
without additional empirical measurements, it is unclear whether this decrease was caused by
bacterial methyl transferase enzyme activity or by photochemical degradation at higher $pCO_2$.
Author contribution: Gui-peng Yang and Kun-shan Gao designed the experiments. Sheng-hui
Zhang and Qiong-yao Ding carried out the experiments and prepared the manuscript. Hong-hai
Zhang and Da-wei Pan revised the paper.
**Acknowledgements**
This study was financially supported by the National Natural Science Foundation of China (Grant
Nos. 41320104008 and 41576073), the National Key Research and Development Program of
China (Grant No. 2016YFA0601301), the National Natural Science Foundation for Creative





Research Groups (Grant No. 41521064), and AoShan Talents Program of Qingdao National
Laboratory for Marine Science and Technology (No. 2015 ASTP). We are thankful to Minhan Dai
for the nutrient data and to Bangqin Huang for the bacterial data.
Competing interests: The authors declare that they have no conflict of interest.

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







**Figure captions**

Fig. 1. $CO_2$ partial pressure ($p$CO$_2$) and mean chlorophyll $a$ (Chl $a$) concentrations in the HC
(1,000 μatm, solid squares) and LC (400 μatm, white squares) mesocosms (3,000 L).
Fig. 2. Temporal changes of *Thalassiosira weissflogii*, *Phaeodactylum tricornutum* and *Emiliania*
*huxleyi* cell concentrations in the HC (1,000 μatm, solid squares) and LC (400 μatm, white
squares) mesocosms (3,000 L).
Fig. 3 Temporal changes in DMS, DMSP, CHBrCl$_2$, CH$_3$Br, CH$_2$Br$_2$ and CH$_3$I concentrations in
the HC (1,000 μatm, black squares) and LC (400 μatm, white squares) mesocosms (3,000 L).




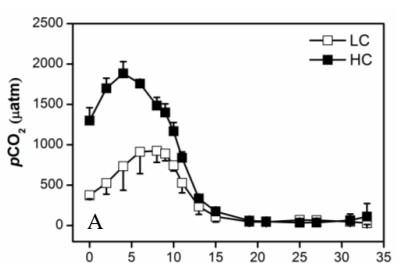


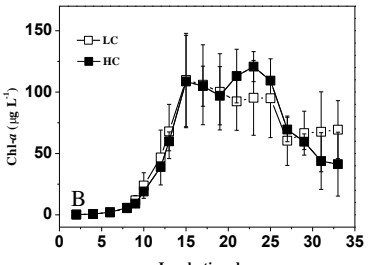


**Fig. 1.** $CO_2$ partial pressure ($pCO_2$) and mean chlorophyll *a* (Chl *a*) concentrations in the HC (1,000 μatm, solid

squares) and LC (400 μatm, white squares) mesocosms (3,000 L). Data are mean ± standard deviation, n = 3

(triplicate independent mesocosm bags) (Origin 8.0).








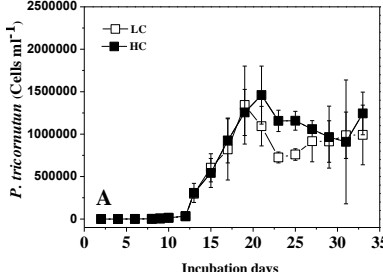


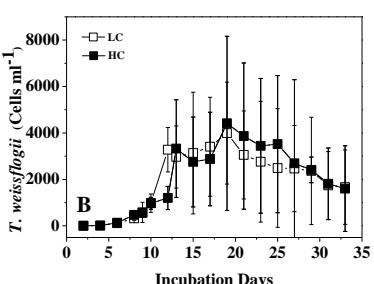

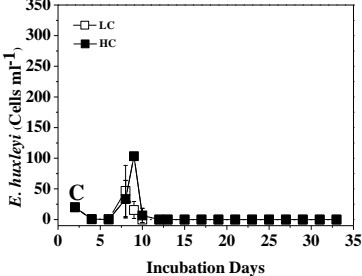


**Fig. 2.** (A) *Thalassiosira weissflogii* cell concentrations; (B) *Phaeodactylum tricornutum* cell concentrations; (C)
*Emiliania huxleyi* cell concentrations. White squares represent the LC (400 µatm) treatment. Data are mean ±
standard deviation, n = 3 (triplicate independent mesocosm bags) (Origin 8.0).






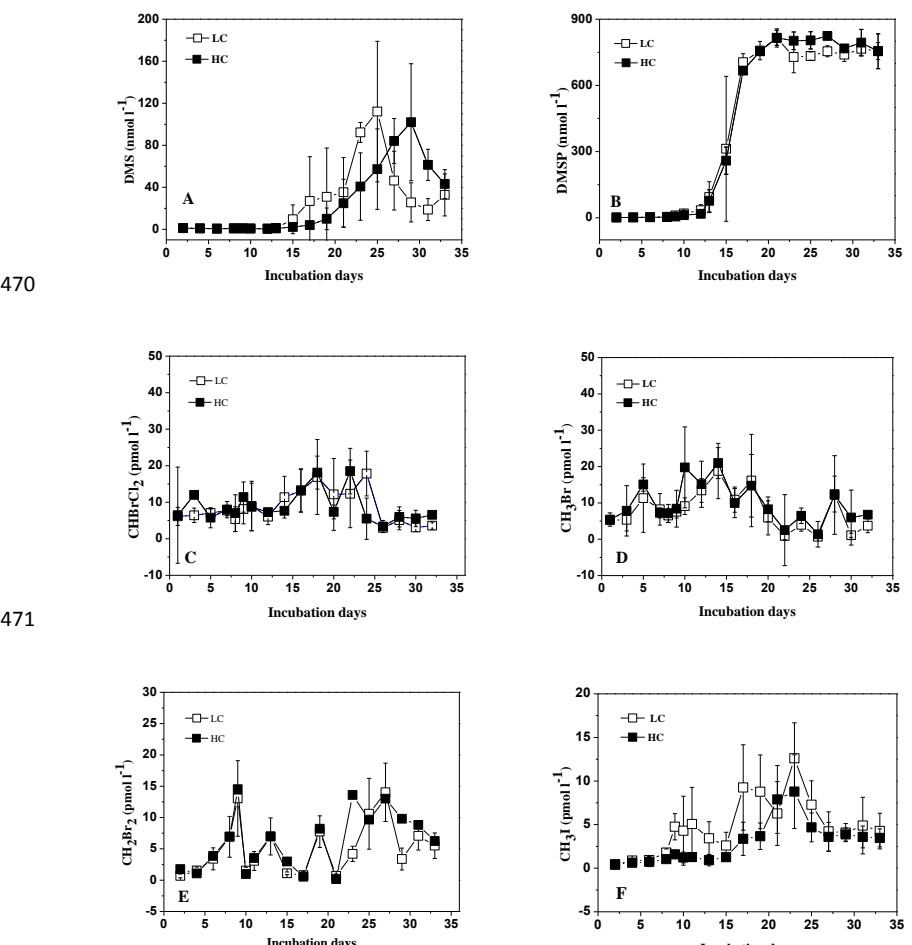




**Fig. 3** Temporal changes in DMS, DMSP, CHBrCl$_2$, CH$_3$Br, CH$_2$Br$_2$ and CH$_3$I concentrations in the HC (1,000

μatm, black squares) and LC (400 μatm, white squares) mesocosms (3,000 L). Data are mean ± standard deviation,

n = 3 (triplicate independent mesocosm bags) (Origin 8.0).





**Table 1**. The conditions of DIC, $pH_T$, $pCO_2$ and nutrient concentrations in the mesocosm
experiments. "-" means that the values were below the detection limit.

| | | $pH_T$ | DIC ($\mu mol\ kg^{-1}$) | $pCO_2$ ($\mu atm$) | $NO_3^-+NO_2^-$ ($\mu mol\ L^{-1}$) | $NH_4^+$ ($\mu mol\ L^{-1}$) | $PO_4^{3-}$ ($\mu mol\ L^{-1}$) | $SiO_3^{2-}$ ($\mu mol\ L^{-1}$) |
|---|---|---|---|---|---|---|---|---|
| day 0 | LC | 8.0±0.1 | 2181±29 | 1170~1284 | 52~56 | 19~23 | 2.6±0.2 | 38~40 |
| | HC | 7.5±0.1 | 2333±34 | 340~413 | 51~55 | 19~23 | 2.5±0.2 | 38~39 |
| PhaseI | LC | 7.9~8.4 | 1825~2178 | 373~888 | 15~52 | 1.6~20 | 0.5~2.6 | 31~38 |
| | HC | 7.4~8.2 | 2029~2338 | 1295~1396 | 47~54 | 0.2~21 | 0.7~2.5 | 34~39 |
| Phase II | LC | 8.4~8.5 | 1706~1745 | 46~749 | -~ 15.9 | - | 0.1~0.5 | 10~24 |
| | HC | 8.4~8.6 | 1740~1891 | 59~1164 | 1.1~25 | - | -~0.1 | 29~30 |
| Phase III | LC | 8.5~8.8 | 1673~1706 | 30~43 | - | - | - | 10~16 |
| | HC | 8.6~8.7 | 1616~1740 | 34~110 | - | - | -~0.3 | 24~25 |


