# Peer review of "Effect of elevated $pCO_2$ on trace gas production during an"

_Biogeosciences, 2018_

## Referee Comment (RC1) · B. Qu (Referee) · 27 May 2018

Increases of anthropogenic emissions of CO2 since the Industrial Revolution are known to have influenced organisms and the delivery of oceanic ecosystem services at a global scale. This is an interesting piece of work that shows the effect of elevated pCO2 on trace gases production including DMS and four halocarbon compounds through a mesocosm experiment. The study is based on the development of a bloom created by the addition of three different species of cultured phytoplankton to nutrient enriched coastal water enclosed in mesocosms. Considering that the impact of ocean acidification on DMS and halocarbons remains controversial, it is necessary to conduct

further study about this aspect. Overall, this paper is well written and the major points are discussed with clarity. I recommend this article to be published in Biogeosciences after modification. My major criticism to the manuscript is that the authors point the algae and their attached bacteria in the coastal environment were removed through filtration process, have you measured the bacterial abundance in the mesocosm before the three different species of algae inoculated? In addition, this manuscript lacks the initial concentrations of Phaeodactylum tricornuntum, Thalassiosira weissflogii, and Emiliania huxleyi inoculated into the mesocosm. There are also some minor thinks that I list below: P3, L54 "Further decreases of 0.3–0.4 pH units are predicted by the end of this century (Doneyet al., 2009; Orr et al., 2005), which is commonly referred to as ocean acidification (OA)." Please update the latest references in this section. P3, L61 "DMS is the most important volatile sulfur compound produced from the algal secondary metabolite dimethylsulfoniopropionate (DMSP) through complex biological interactions in marine ecosystems (Stefels et al., 2007)." DMSP is not only produced by algae, but also by terrestrial plants and marine bacteria. Please re-word this section. P4, L75 Replace "attribute" by "attributed". P8, L167-L168 What is "LC" and "HC", low CO2 and high CO2? Please use the full name for the first time in the manuscript. P8, L172 The unit of chl a is not unified with Fig. 1, please check. P9, L192 Replace "for" by "of" P9, L196 delete "growth in" P9, L197-198 Replace "increase in Chl a and cell concentrations" by "increase in Chl a concentrations and algal cells"
* * *

---

## Referee Comment (RC2) · Anonymous Referee #2 · 18 Jun 2018

Title: Effect of elevated pCO2 on trace gas production during an ocean acidification mesocosm experiment Author(s): Sheng-Hui Zhang et al. MS No.: bg-2018-148

General Comments

The study examines production of volatile sulfur and halocarbon compounds in meso-cosms of seawater with different dissolved carbon concentrations. The premise is to examine the impact of ocean acidification on gas production.

This is an okay idea. One major concern, however, is that the study was only five-weeks long, and there was no pretreatment of the phytoplankton. Thus, it is not really a global change test, but rather it is a test of acid shock on phytoplankton. I suppose

this is interesting.

Also, it appears to me that some of the data on temporal changes in chemistry and biology in the mesocosms have been published previously by Liu et al. (2017). Figure 1 is identical to Figure 1 and Figure 2 in Liu et al. (2107) and, at least, two panels in Figure 2 are in Figure 3 in Liu et al. (2017).

Thus, only the data in Figure 3 are new. Unfortunately, you cannot publish the same data twice. Elsevier, the publisher of Marine Environmental Research, owns the copyright to those figures.

Specific Comments

1) The abstract reads well.

2) The introduction is okay. However, it ends a bit abruptly. As written it is mostly a review of literature ending in an objective to do more research. Although a research objective is good, research should be question driven and present a testable, falsifiable hypothesis. In this case, what do you hope to learn in a 5-week study? (This seems short term to me.)

3) The methods seem appropriate, to me.

4) The results are okay. However, the discussion about the role of bacteria in DMSP dynamics, on page 10 and 11, seems like speculation to me. Where are the data on bacteria in the mesocosms? Speculation is okay, but data is better.

5) Than many correlations in the text could go in a table. This would make the text more readable.

6) Much of the discussion on page 13 is literature rather than interpretation. Rather than merely list other studies, compare results quantitatively. Did the other studies have CH3I production rates that were similar to yours?

Technical Comments

1) Line 31 & 36: report the percentages as whole integers. It is nearly impossible to measure accurately to 0.1%.

2) Line 48: 'human activity' and 'anthropogenic' are the same. You do not need both in the sentence.

3) Line 69: delete the sentence 'several studies have already, etc.' in the following sentence, replace 'majority' with 'several studies have shown a negative impact, etc.'

4) Line 78: perhaps start a new paragraph with 'halocarbons'

5) Line 189 to 192: delete. This is not an appropriate topic sentence, and it is from the introduction. No need to repeat here.

6) Line 192: delete the sentence and put (Fig. 3) in the following sentence.

7) Line 209: round '29.2%' to the '29%'.

8) Line 228: why Yu et al., unpublished data? Why not include the data here?

9) Line 258: the sentence does not make sense. Do you mean 'attributed to biology' rather than 'involve'. Also delete the quotes around 'biogenic'. Why use quotes for an adjective?

Sorry but I cannot overlook the attempt to publish the same data in two papers. I realize that data from one paper can be used in another, but this needs to acknowledge the first paper and copywrite.

---

## Author Comment (AC1) · 29 Jul 2018

Dear Reviewer #1: We are grateful to your review of this paper and would like to express our thanks for your helpful and constructive comments. We have revised the manuscript and addressed all the comments point by point. The main changes we made are as follows: Increases of anthropogenic emissions of CO2 since the Industrial Revolution are known to have influenced organisms and the delivery of oceanic ecosystem services at a global scale. This is an interesting piece of work that shows the effect of elevated pCO2 on trace gases production including DMS and four halocarbon compounds through a mesocosm experiment. The study is based on the development of a bloom created by the addition of three different species of cultured phytoplankton to nutrient enriched coastal water enclosed in mesocosms. Considering that the impact of ocean acidification on DMS and halocarbons remains controversial, it is necessary to conductfurther study about this aspect. Overall, this paper is well written and the major points are discussed with clarity. I recommend this article to be published in Biogeosciences after modification. My major criticism to the manuscript is that the authors point the algae and their attached bacteria in the coastal environment were removed through filtration process, have you measured the bacterial abundance in the mesocosm before the three different species of algae inoculated? In addition, this manuscript lacks the initial concentrations of Phaeodactylum tricornuntum, Thalassiosira weissflogii, and Emiliania huxleyi inoculated into the mesocosm. Thanks for the reviewer's suggestion and we have added some details about this mesocosm experiment in the revised manuscript. P6, L125-129 "Emiliania huxleyi (CS-369), Phaeodactylum tricornuntum (CCMA 106), and Thalassiosira weissflogii (CCMA 102) were inoculated into the mesocosm bags, with initial diatom/coccolithophorid cell ratio was 1:1. The initial concentrations of Phaeodactylum tricornuntum, Thalassiosira weissflogii, and Emiliania huxleyi inoculated into the mesocosm were 10, 10, and 20 cells mL$-1$, respectively." P7, L141-142 "Meanwhile, no meaningful numbers of bacteria were counted by flow cytometer in the pre-filtered seawater before the inoculations." There are also some minor thinks that I list below: P3, L54 "Further decreases of 0.3–0.4 pH units are predicted by the end of this century (Doneyet al., 2009; Orr et al., 2005), which is commonly referred to as ocean acidification (OA)." Please update the latest references in this section. Thanks for the reviewer's suggestion and we have updated the latest references in the revised manuscript. P3, L58-60 "Further decreases of 0.3–0.4 pH units are predicted by the end of this century (Doney et al., 2009; Orr et al., 2005; Gattuso et al., 2015), which is commonly referred to as ocean acidification (OA)" "Gattuso, J. P., Magnan, A., Bille, R., Cheung, W. W. L., Howes, E. L., Joos, F., Allemand, D., Bopp, L., Cooley, S. R., Eakin, C. M., Hoegh-Guldberg, O., Kelly, R. P., Portner, H. O., Rogers, A. D., Baxter, J. M., Laffoley, D., Osborn, D.,

Rankovic, A., Rochette, J., Sumaila, U.R., Treyer, S., Turley, C.: Contrasting futures for ocean and society from different anthropogenic CO2 emissions scenarios. Science, 349 (6243), aac4722, 2015." P3, L61 "DMS is the most important volatile sulfur compound produced from the algal secondary metabolite dimethylsulfoniopropionate (DMSP) through complex biological in teractions in marine ecosystems (Stefels et al., 2007)." DMSP is not only produced by algae, but also by terrestrial plants and marine bacteria. Please re-word this section. Thanks for the reviewer's suggestion and we have reworded this section in the revised manuscript. P3, L67-71 "DMS is the most important volatile sulfur compound produced from dimethylsulfoniopropionate (DMSP), which is ubiquitous in marine environments, mainly synthesized by marine microalgae (Stefels et al., 2007), a few angiosperms, some corals (Raina et al., 2016), and several heterotrophic bacteria (Curson et al., 2017) through complex biological interactions in marine ecosystems." "Raina, J. B., Tapiolas, D., Motti, C. A., Foret, S., Seemann, T., Tebben, J.: Isolation of an antimicrobial compound produced by bacteria associated with reef-building corals. PeerJ, 4, e2275, 2016" "Curson, A. R., Liu, J., Bermejo Martinez, A., Green, R., Chan, Y., Carrion, O.: Dimethylsulfoniopropionate biosynthesis in marine bacteria and identification of the key gene in this process. Nat. Microbiol., 2, 17009, 2017." P4, L75 Replace "attribute" by "attributed". Thanks for the reviewer's suggestion and we have reworded this section in the revised manuscript. P4, L80-84 "Several assumptions have been presented to explain these contrasting results and attributed the pH-induced variation in DMS-production capability to altered physiology of the algae cells or of bacterial DMSP degradation (Vogt et al., 2008; Hopkins et al., 2010, Avgoustidi et al., 2012; Archer et al., 2013; Hopkins and Archer, 2014; Webb et al., 2015)" P8, L167-L168 What is "LC" and "HC", low CO2 and high CO2? Please use the full name for the first time in the manuscript. Thanks for the reviewer's suggestion and we have used the full name for the first time in the revised manuscript. P9, L192-195 "The initial chemical parameters of the mesocosm experiment are shown in Table 1. The initial mean dissolved nitrate (including NO3‒ and NO2‒), NH4+, PO43‒ and silicate (SiO32‒) concentrations were 54, 20, 2.6 and 41 $\mu$mol L‒1

[Figure]

for the low pCO2 (LC) treatment and 52, 21, 2.4 and 38 $\mu$mol L‒1 for the high pCO2 (HC) treatment, respectively." P8, L172 The unit of chl a is not unifi̧ed with Fig. 1, please check. According to the opinion of reviewer 2#, Fig. 1 was replaced.

Fig. 1. Temporal changes of pH in the HC (1,000 $\mu$atm, solid squares) and LC (400 $\mu$atm, white squares) mesocosms (3,000 L). Data are mean $\pm$ standard deviation, n = 3 (triplicate independent mesocosm bags) (Origin 8.0). P9, L192 Replace "for" by "of" Thanks for the reviewer's suggestion and we have reworded in the revised manuscript according all reviewers' suggestion. P10, L207-L209 "At the beginning of the experiment, the mean DMS, DMSP and DCB concentrations were all low in both treatments due to the low concentrations of DMS, DMSP and DCB in the original fjord water and possible loss during the filtration procedure (Fig. 2)." P9, L196 delete "growth in" Thanks for the reviewer's suggestion and we have modified in the revised manuscript. P10, L217-218 "Compared with DMSP, DMS and DCB concentrations showed similar trends during the mesocosm experiment." P9, L197-198 Replace "increase in Chl a and cell concentrations" by "increase in Chl a concentrations and algal cells" Thanks for the reviewer's suggestion and we have modified in the revised manuscript. P10, L210-212 "The DMSP concentrations in the HC and LC treatments increased significantly along with the increase of Chl a concentrations and algal cells, and stayed relatively constant over the following days."

Please also note the supplement to this comment:
https://www.biogeosciences-discuss.net/bg-2018-148/bg-2018-148-AC1-supplement.pdf
* * *
[Figure]

**Supplement:**

B. Qu (Referee)

2467327342@qq.com

Increases of anthropogenic emissions of CO$_2$ since the Industrial Revolution are known to have influenced organisms and the delivery of oceanic ecosystem services at a global scale. This is an interesting piece of work that shows the effect of elevated $p$CO$_2$ on trace gases production including DMS and four halocarbon compounds through a mesocosm experiment. The study is based on the development of a bloom created by the addition of three different species of cultured phytoplankton to nutrient enriched coastal water enclosed in mesocosms. Considering that the impact of ocean acidification on DMS and halocarbons remains controversial, it is necessary to conductfurther study about this aspect. Overall, this paper is well written and the major points are discussed with clarity. I recommend this article to be published in Biogeosciences after modification. My major criticism to the manuscript is that the authors point the algae and their attached bacteria in the coastal environment were removed through filtration process, have you measured the bacterial abundance in the mesocosm before the three different species of algae inoculated? In addition, this manuscript lacks the initial concentrations of Phaeodactylum tricornuntum, Thalassiosira weissflogii, and Emiliania huxleyi inoculated into the mesocosm.

There are also some minor thinks that I list below:

P3, L54 "Further decreases of 0.3–0.4 pH units are predicted by the end of this century (Doneyet al., 2009; Orr et al., 2005), which is commonly referred to as ocean acidification (OA)." Please update the latest references in this section.

P3, L61 "DMS is the most important volatile sulfur compound produced from the algal secondary metabolite dimethylsulfoniopropionate (DMSP) through complex biological in teractions in marine ecosystems (Stefels et al., 2007)." DMSP is not only produced by algae, but also by terrestrial plants and marine bacteria. Please re-word this section.

P4, L75 Replace "attribute" by "attributed".

P8, L167-L168 What is "LC" and "HC", low CO$_2$ and high CO$_2$? Please use the full name for the first time in the manuscript.

P8, L172 The unit of chl a is not unified with Fig. 1, please check.

P9, L192 Replace "for" by "of"

P9, L196 delete "growth in"

P9, L197-198 Replace "increase in Chl $a$ and cell concentrations" by "increase in Chl $a$

concentrations and algal cells"

Response to Reviewer #1:

Dear Reviewer #1:

We are grateful to your review of this paper and would like to express our thanks for your helpful and constructive comments. We have revised the manuscript and addressed all the comments point by point. The main changes we made are as follows:

Increases of anthropogenic emissions of $CO_2$ since the Industrial Revolution are known to have influenced organisms and the delivery of oceanic ecosystem services at a global scale. This is an interesting piece of work that shows the effect of elevated $p CO_2$ on trace gases production including DMS and four halocarbon compounds through a mesocosm experiment. The study is based on the development of a bloom created by the addition of three different species of cultured phytoplankton to nutrient enriched coastal water enclosed in mesocosms. Considering that the impact of ocean acidification on DMS and halocarbons remains controversial, it is necessary to conductfurther study about this aspect. Overall, this paper is well written and the major points are discussed with clarity. I recommend this article to be published in Biogeosciences after modification. My major criticism to the manuscript is that the authors point the algae and their attached bacteria in the coastal environment were removed through filtration process, have you measured the bacterial abundance in the mesocosm before the three different species of algae inoculated? In addition, this manuscript lacks the initial concentrations of *Phaeodactylum*

*tricornuntum*, *Thalassiosira weissflogii*, and *Emiliania huxleyi* inoculated into the mesocosm.

Thanks for the reviewer's suggestion and we have added some details about this mesocosm experiment in the revised manuscript.

P6, L125-129 "*Emiliania huxleyi* (CS-369), *Phaeodactylum tricornuntum* (CCMA 106), and

*Thalassiosira weissflogii* (CCMA 102) were inoculated into the mesocosm bags, with initial diatom/coccolithophorid cell ratio was 1:1. The initial concentrations of *Phaeodactylum*

*tricornuntum*, *Thalassiosira weissflogii*, and *Emiliania huxleyi* inoculated into the mesocosm were

10, 10, and 20 cells $mL^{-1}$, respectively."

P7, L141-142 "Meanwhile, no meaningful numbers of bacteria were counted by flow cytometer in the pre-filtered seawater before the inoculations."

There are also some minor thinks that I list below:

P3, L54 "Further decreases of 0.3–0.4 pH units are predicted by the end of this century (Doneyet al., 2009; Orr et al., 2005), which is commonly referred to as ocean acidification (OA)." Please update the latest references in this section.

Thanks for the reviewer's suggestion and we have updated the latest references in the revised manuscript.

P3, L58-60 "Further decreases of 0.3–0.4 pH units are predicted by the end of this century (Doney et al., 2009; Orr et al., 2005; Gattuso et al., 2015), which is commonly referred to as ocean acidification (OA)"

"Gattuso, J. P., Magnan, A., Bille, R., Cheung, W. W. L., Howes, E. L., Joos, F., Allemand, D.,

Bopp, L., Cooley, S. R., Eakin, C. M., Hoegh-Guldberg, O., Kelly, R. P., Portner, H. O.,

Rogers, A. D., Baxter, J. M., Laffoley, D., Osborn, D., Rankovic, A., Rochette, J., Sumaila,

U.R., Treyer, S., Turley, C.: Contrasting futures for ocean and society from different anthropogenic $CO_2$ emissions scenarios. Science, 349 (6243), aac4722, 2015."

P3, L61 "DMS is the most important volatile sulfur compound produced from the algal secondary metabolite dimethylsulfoniopropionate (DMSP) through complex biological in teractions in marine ecosystems (Stefels et al., 2007)." DMSP is not only produced by algae, but also by terrestrial plants and marine bacteria. Please re-word this section.

Thanks for the reviewer's suggestion and we have reworded this section in the revised manuscript.

P3, L67-71 "DMS is the most important volatile sulfur compound produced from dimethylsulfoniopropionate (DMSP), which is ubiquitous in marine environments, mainly synthesized by marine microalgae (Stefels et al., 2007), a few angiosperms, some corals (Raina et al., 2016), and several heterotrophic bacteria (Curson et al., 2017) through complex biological interactions in marine ecosystems."

"Raina, J. B., Tapiolas, D., Motti, C. A., Foret, S., Seemann, T., Tebben, J.: Isolation of an antimicrobial compound produced by bacteria associated with reef-building corals. PeerJ, 4, e2275, 2016"

"Curson, A. R., Liu, J., Bermejo Martinez, A., Green, R., Chan, Y., Carrion, O.:

Dimethylsulfoniopropionate biosynthesis in marine bacteria and identification of the key gene in this process. Nat. Microbiol., 2, 17009, 2017."

P4, L75 Replace "attribute" by "attributed".

Thanks for the reviewer's suggestion and we have reworded this section in the revised manuscript.

P4, L80-84 "Several assumptions have been presented to explain these contrasting results and attributed the pH-induced variation in DMS-production capability to altered physiology of the algae cells or of bacterial DMSP degradation (Vogt et al., 2008; Hopkins et al., 2010, Avgoustidi et al., 2012; Archer et al., 2013; Hopkins and Archer, 2014; Webb et al., 2015)"

P8, L167-L168 What is "LC" and "HC", low $CO_2$ and high $CO_2$? Please use the full name for the first time in the manuscript.

Thanks for the reviewer's suggestion and we have used the full name for the first time in the revised manuscript.

P9, L192-195 "The initial chemical parameters of the mesocosm experiment are shown in Table 1.

The initial mean dissolved nitrate (including $NO_3^-$ and $NO_2^-$), $NH_4^+$, $PO_4^{3-}$ and silicate ($SiO_3^{2-}$)

concentrations were 54, 20, 2.6 and 41 $\mu$mol $L^{-1}$ for the low $p$CO$_2$ (LC) treatment and 52, 21, 2.4

and 38 $\mu$mol $L^{-1}$ for the high $p$CO$_2$ (HC) treatment, respectively."

P8, L172 The unit of chl $a$ is not unified with Fig. 1, please check.

According to the opinion of reviewer 2#, Fig. 1 was replaced.

[Figure]

**Fig. 1.** Temporal changes of pH in the HC (1,000 µatm, solid squares) and LC (400 µatm, white squares) mesocosms (3,000 L). Data are mean ± standard deviation, n = 3 (triplicate independent mesocosm bags) (Origin 8.0).

P9, L192 Replace "for" by "of"

Thanks for the reviewer's suggestion and we have reworded in the revised manuscript according all reviewers' suggestion.

P10, L207-L209 "At the beginning of the experiment, the mean DMS, DMSP and DCB concentrations were all low in both treatments due to the low concentrations of DMS, DMSP and DCB in the original fjord water and possible loss during the filtration procedure (Fig. 2)."

P9, L196 delete "growth in"

Thanks for the reviewer's suggestion and we have modified in the revised manuscript.

P10, L217-218 "Compared with DMSP, DMS and DCB concentrations showed similar trends during the mesocosm experiment."

P9, L197-198 Replace "increase in Chl *a* and cell concentrations" by "increase in Chl *a* concentrations and algal cells"

Thanks for the reviewer's suggestion and we have modified in the revised manuscript.

[revised manuscript text omitted]
* | 0.635** | 0.954** | 0.803** | 0.143 | -0.257 | 0.267 | 0.834** | 0.559 | 0.820** | 1 |

---

## Author Comment (AC2) · 29 Jul 2018

Dear Reviewer #2: We are grateful to your review of this paper and would like to express our thanks for the helpful and constructive comments. We have revised the manuscript and addressed all the comments. The main changes we made are as follows: General Comments The study examines production of volatile sulfur and halo-carbon compounds in mesocosms of seawater with different dissolved carbon concentrations. The premise is to examine the impact of ocean acidification on gas production. This is an okay idea. One major concern, however, is that the study was only five-weeks long, and there was no pretreatment of the phytoplankton. Thus, it is not really a global change test, but rather it is a test of acid shock on phytoplankton. I suppose this is interesting. Thanks for the reviewer's suggestion and we also agree with that the mesocosm experiment is a test of acid shock on phytoplankton. In addition, simple pretreatment was conducted before the mesocosm experiment as described in Huang et al. (2018). Briefly, before being introduced into the mesocosms, the three phytoplankton species and their associated bacteria were cultured in autoclaved, pre-filtered seawater from Wuyuan Bay at 16°C (similar to the in situ temperature of Wuyuan Bay) without any addition of nutrients. Cultures were continuously aerated with filtered ambient air containing 400 $\mu$atm of CO2 within plant chambers (HP1000G-D, Wuhan Ruihua Instrument & Equipment, China) at a constant bubbling rate of 300 ml min–1. The culture medium was renewed every 24 h to maintain the cells of each phytoplankton species in exponential growth. We have added these pretreatment in the revised manuscript. P6, L135-P7, L141 "Before being introduced into the mesocosms, the three phytoplankton species were cultured in autoclaved, pre-filtered seawater from Wuyuan Bay at 16 °C (similar to the in situ temperature of Wuyuan Bay) without any addition of nutrients. Cultures were continuously aerated with filtered ambient air containing 400 $\mu$atm of CO2 within plant chambers (HP1000G-D, Wuhan Ruihua Instrument & Equipment, China) at a constant bubbling rate of 300 mL min–1. The culture medium was renewed every 24 h to maintain the cells of each phytoplankton species in exponential growth." Also, it appears to me that some of the data on temporal changes in chemistry and biology in the mesocosms have been published previously by Liu et al. (2017). Figure 1 is identical to Figure 1 and Figure 2 in Liu et al. (2017) and, at least, two panels in Figure 2 are in Figure 3 in Liu et al. (2017). Thus, only the data in Figure 3 are new. Unfortunately, you cannot publish the same data twice. Elsevier, the publisher of Marine Environmental Research, owns the copyright to those figures. We agree with the reviewer's suggestion. In the revised manuscript, we deleted the conflicting figures and only described these results simply. P9, L187-L192"The phytoplankton growth process was divided into three phases in terms of variations in Chl a concentrations in the mesocosm experiments as described in Liu et al. (2017): i) the logarithmic growth phase (phase I, days 0–13), ii) a plateau phase (phase II, days 13–23, bloom period), and iii) a secondary plateau phase (phase III, days 23–33) attained after a decline in biomass from a maximum in phase II." P9, L200-L204 "Emiliania huxleyi was only found in phase I and its maximal concentration reached 310 cells mL‒1 according to the results of microscopic inspection. Thalassiosira weissflogii was found throughout the entire period in each bag, but the maximum concentration was 8,120 cells mL‒1, which was far less than the concentration of Phaeodactylum tricornutum with a maximum density of about 1.5 million cells mL‒1 (Liu et al., 2017)." Specific Comments 1) The abstract reads well. Thanks for the reviewer's ratification. 2) The introduction is okay. However, it ends a bit abruptly. As written it is mostly a review of literature ending in an objective to do more research. Although a research objective is good, research should be question driven and present a testable, falsifiable hypothesis. In this case, what do you hope to learn in a 5-week study? (This seems short term to me.) We agree with the reviewer's suggestion and have made modification in the revised manuscript. P5, L97-103 "DMS and halocarbons play a significant role in the global climate and perhaps act a greater extent in the future. Meanwhile, the combined picture arising from existing studies is that the response of communities to OA is not predictable and further studies were required. Based on the controversial results about OA on DMS and halocarbons production, a mesocosm experiment was conducted in Wu Yuan Bay, Xiamen. The aim of this study was to investigate the influence of elevated $pCO_2$ on diatoms and coccolithophores and to further understand how the productions of DMS and halocarbons respond to OA." During this experiment, the nutrient concentrations (dissolved inorganic nitrogen (DIN) and phosphate) in phase II were below or close to the detection limit, though the Chl a concentration still maintained a relatively high concentration after 5 weeks incubation. We think that the stored nutrients in diatom cells might contribute to the biomass increase even after the depletion of nutrients in the surrounding seawater (Goldman et al., 1979; Sommer, 1989). Meanwhile, DMS, DCB and CH3I concentration decreased significantly after 5 weeks incubation. Therefore, 5 weeks incubation is appropriate to this experiment. 3) The methods seem appropriate, to me. Thanks for the reviewer's ratification. 4) The results are okay. However, the discussion about the role of bacteria in DMSP dynamics, on page 10 and 11, seems like speculation to me. Where are the data on bacteria in the mesocosms? Speculation is okay, but data is better. We agree with the reviewer's suggestion. We have added the DCB data in the revised manuscript. P8, L167-L175 "2.5 Enumeration of DMSP-consuming bacteria (DCB) The number of DMSP-consuming bacteria (DCB) was estimated using the most probable number (MPN) methodology. The MPN medium consisted of a mixture (1:1 v/v) of sterile artificial sea water (ASW) and mineral medium (Visscher et al., 1991), 3 mL of which was dispensed in 6 mL test tubes, which were closed off by an over-sized cap, allowing gas exchange. Triplicate dilution series were set up. All test tubes contained 1 mmol L-1 DMSP as the sole organic carbon source and were kept at 30 °C in the dark. After 2 weeks, the presence/absence of bacteria in the tubes was verified by DAPI staining (Porter and Feig, 1980). Three tubes containing 3 mL ASW without substrate were used as controls." P10, L217-L224 "Compared with DMSP, DMS and DCB concentrations showed similar trends during the mesocosm experiment. DMS concentrations in the LC and HC treatments were 1.03 and 0.74 nmol L−1, respectively, while DCB concentrations in the LC and HC treatments were 0.20 × 106 and 0.16 × 106 cells mL−1. DMS and DCB concentrations did not increase significantly during phase I, but began to increase rapidly on day 15. DCB concentrations in the LC and HC treatments peaked on days 21 (11.65 × 106 cells mL−1) and 23 (10.70 × 106 cells mL−1), while DMS concentrations in the LC and HC treatments peaked on days 25 (112.1 nmol L−1) and 30 (101.9 nmol L−1). Both DMS and DCB concentrations began to decrease obviously during phase III." P11, L231-L234 "However, a significant 29% reduction in DMS concentrations was detected in the HC treatment compared with the LC treatment (p = 0.016), though no statistical difference for DCB concentrations was found between the LC and HC treatments during phase I." P11, L244-L246 "In addition, a significant positive relationship was also observed between DMS and DCB (r = 0.643, p < 0.01 in the LC treatment; r = 0.544, p < 0.01 in the HC treatment) during this experiment." P12, L251-L253 "Moreover, DCB peaked on days 21 (11.65 × 106 cells mL-1) and 23 (10.70 × 106 cells mL-1) in the LC and HC treatments, respectively, as shown in Fig. 2-C. Similar to DMS, DCB was also delayed in the HC mesocosm compared to that in the LC mesocosm." 5) Than many correlations in the text could go in a table. This would make the text more readable. We agree with reviewer's suggestion and have add two tables in the revised manuscript.

Table 2. Relationships between DMS, DMSP, Chl a, CHBrCl2, CH3Br, CH2Br2, CH3I, DCB, Thalassiosira weissflogii (T. weissflogii) and Phaeodactylum tricornutum (P. tricornutum) concentrations in the LC treatments. DMS (nmol L-1) DMSP (nmol L-1) Chl a (μg L-1) CHBrCl2 (pmol l-1) CH3Br (pmol l-1) CH2Br2 (pmol l-1) CH3I (pmol l-1) DCB (×106 cells mL-1) T. weissflogii (×103 cells mL-1) P. tricornutum (cells mL-1) DMS 1 DMSP 0.701** 1 Chl a 0.597** 0.792** 1 CHBrCl2 0.526 0.280 0.559 1 CH3Br -0.413 -0.230 0.196 0.313 1 CH2Br2 0.310 0.180 0.001 -0.136 -0.308 1 CH3I 0.694** 0.654** 0.717** 0.596* -0.151 0.129 1 DCB 0.643** 0.520* 0.522* 0.394 -0.268 -0.038 0.762** 1 T. weissflogii 0.410 0.617** 0.899** 0.301 0.322 0.028 0.680** 0.399 1 P. tricornutum 0.560* 0.961** 0.821** 0.528 -0.032 0.162 0.588** 0.334 0.685** 1

Table 3. Relationships between DMS, DMSP, Chl a, CHBrCl2, CH3Br, CH2Br2, CH3I, DCB, Thalassiosira weissflogii (T. weissflogii) and Phaeodactylum tricornutum (P. tricornutum) concentrations in the HC treatments. DMS (nmol L-1) DMSP (nmol L-1) Chl a (μg L-1) CHBrCl2 (pmol l-1) CH3Br (pmol l-1) CH2Br2 (pmol l-1) CH3I (pmol l-1) DCB (×106 cells mL-1) T. weissflogii (×103 cells mL-1) P. tricornutum (cells mL-1) DMS 1 DMSP 0.752** 1 Chl a 0.318* 0.738** 1 CHBrCl2 0.324 0.094 0.326 1 CH3Br -0.410 -0.349 0.065 0.076 1 CH2Br2 0.540* 0.352 0.142 0.233 -0.377 1 CH3I 0.694** 0.816** 0.741** 0.690* -0.407 0.316 1 DCB 0.544* 0.522 0.549* 0.532 -0.311 0.368 0.851* 1 T. weissflogii 0.355 0.743** 0.930** 0.304 0.076 0.233 0.690** 0.567 1 P. tricornutum 0.635** 0.954** 0.803** 0.143 -0.257 0.267 0.834** 0.559 0.820** 1

6) Much of the discussion on page 13 is literature rather than interpretation. Rather than merely list other studies, compare results quantitatively. Did the other studies have CH3I production rates that were similar to yours? We agree with reviewer's suggestion and have made the modification in the revised manuscript. P13, L280-288 "The temporal dynamics of CH3I in the HC and LC treatments are shown in Fig. 3-D. The CH3I concentrations in the LC treatment varied from 0.38 to 12.61 pmol L-1, with a mean of 4.76 pmol L-1. The CH3I concentrations in the HC treatment ranged between 0.44 and 8.78 pmol L-1, with a mean of 2.88 pmol L-1. The maximum CH3I concentrations in the HC and LC treatments were both observed on day 23. The range of CH3I concentrations during this experiment was similar to that measured in the mesocosm experiment (< 1∼10 pmol L-1) in Kongsfjorden conducted by Hopkins et al. (2013). In addition, the mean CH3I concentration in the LC treatment was similar to that measured in the East China Sea, with an average of 5.34 pmol L-1 in winter and 5.74 pmol L-1 in summer (Yuan et al., 2015)." Technical Comments 1) Line 31 & 36: report the percentages as whole integers. It is nearly impossible to measure accurately to 0.1%. Thanks for the reviewer's suggestion. We have modified this in the revised manuscript. P2, L36-43 "During the logarithmic growth phase (phase I), DMS concentrations in high pCO2 mesocosms (HC, 1000 $\mu$atm) were 28% lower than those in low pCO2 mesocosms (LC, 400 $\mu$atm). Elevated pCO2 led to a delay in DCB concentrations attached to Thalassiosira weissflogii and Phaeodactylum tricornutum and finally resulted in the delay of DMS concentration in the HC treatment. Unlike DMS, the elevated pCO2 did not affect DMSP production ability of Thalassiosira weissflogii or Phaeodactylum tricornuntum throughout the 5 weeks culture. A positive relationship was detected between CH3I and Thalassiosira weissflogii and Phaeodactylum tricornuntum during the experiment, and there was a 40% reduction in mean CH3I concentrations in the HC mesocosms." 2) Line 48: 'human activity' and 'anthropogenic' are the same. You do not need both in the sentence. We agree with the reviewer's suggestion and have modified this in the revised manuscript. P3, L53-56 "Anthropogenic emissions have increased the fugacity of atmospheric carbon dioxide (pCO2) from the pre-industrial value of 280 $\mu$atm to the present-day value of over 400 $\mu$atm, and these values will further increase to 800–1000 $\mu$atm by the end of this century according to the Intergovernmental Panel on Climate Change (Gattuso et al., 2015)." 3) Line 69: delete the sentence 'several studies have already, etc.' in the following sentence, replace 'majority' with 'several studies have shown a negative impact, etc.' We agree with the reviewer's suggestion and have made modification in the revised manuscript. P4, L77-80 "Several studies have shown a negative impact of decreasing pH on DMS-production capability (Hopkins et al., 2010; Avgoustidi et al., 2012; Archer et al., 2013; Webb et al., 2016), while others have found either no effect or a positive effect (Vogt et al., 2008; Hopkins and Archer, 2014)." 4) Line 78: perhaps start a new paragraph with 'halocarbons' Thanks for the reviewer's suggestion. We have started a new paragraph with 'halocarbons' in the revised manuscript.. 5) Line 189 to 192: delete. This is not an appropriate topic sentence, and it is from the introduction. No need to repeat here. Thanks for the reviewer's suggestion. We have deleted this sentence in the revised manuscript. 6) Line 192: delete the sentence and put (Fig. 3) in the following sentence. We agree with the reviewer's suggestion and have made the modification in the revised manuscript. P9, L207-P10, L209 "At the beginning of the experiment, the mean DMS, DMSP and DCB concentrations were all low in both treatments due to the low concentrations of DMS, DMSP and DCB in the original fjord water and possible loss during the filtration procedure (Fig. 2)." 7) Line 209: round '29.2%' to the '29%'. We agree with the reviewer's suggestion and have made the modification in the revised manuscript. P11, L231-234 "However, a significant 29% reduction in DMS concentrations was detected in the HC treatment compared with the LC treatment (p = 0.016), though no statistical difference for DCB concentrations was found between the LC and HC treatments during phase I." 8) Line 228: why Yu et al., unpublished data? Why not include the data here? We agree with the reviewer's suggestion. We have added the DCB data in the revised manuscript. P8, L167-L175 "2.5 Enumeration of DMSP-consuming bacteria (DCB) The number of DMSP-consuming bacteria (DCB) was estimated using the most probable number (MPN) methodology. The MPN medium consisted of a mixture (1:1 v/v) of sterile artificial sea water (ASW) and mineral medium (Visscher et al., 1991), 3 mL of which was dispensed in 6 mL test tubes, which were closed off by an over-sized cap, allowing gas exchange. Triplicate dilution series were set up. All test tubes contained 1 mmol L-1 DMSP as the sole organic carbon source and were kept at 30 °C in the dark. After 2 weeks, the presence/absence of bacteria in the tubes was verified by DAPI staining (Porter and Feig, 1980). Three tubes containing 3 mL ASW without substrate were used as controls." P10, L217-L224 "Compared with DMSP, DMS and DCB concentrations showed similar trends during the mesocosm experiment. DMS concentrations in the LC and HC treatments were 1.03 and 0.74 nmol L–1, respectively, while DCB concentrations in the LC and HC treatments were $0.20 \times 10^6$ and $0.16 \times 10^6$ cells mL–1. DMS and DCB concentrations did not increase significantly during phase I, but began to increase rapidly on day 15. DCB concentrations in the LC and HC treatments peaked on days 21 ($11.65 \times 10^6$ cells mL–1) and 23 ($10.70 \times 10^6$ cells mL–1), while DMS concentrations in the LC and HC treatments peaked on days 25 (112.1 nmol L–1) and 30 (101.9 nmol L–1). Both DMS and DCB concentrations began to decrease obviously during phase III." P11, L231-L234 "However, a significant 29% reduction in DMS concentrations was detected in the HC treatment compared with the LC treatment (p = 0.016), though no statistical difference for DCB concentrations was found between the LC and HC treatments during phase I." P11, L244-L246 "In addition, a significant positive relationship was also observed between DMS and DCB (r = 0.643, p < 0.01 in the LC treatment; r = 0.544, p < 0.01 in the HC treatment) during this experiment." P12, L250-L253 "Moreover, DCB peaked on days 21 ($11.65 \times 10^6$ cells mL-1) and 23 ($10.70 \times 10^6$ cells mL-1) in the LC and HC treatments, respectively, as shown in Fig. 2-C. Similar to DMS, DCB was also delayed in the HC mesocosm compared to that in the LC mesocosm." 9) Line 258: the sentence does not make sense. Do you mean 'attributed to biology' rather than 'involve'. Also delete the quotes around 'biogenic'. Why use quotes for an adjective? We agree with the reviewer's suggestion and have deleted this sentence in the revised manuscript.

Please also note the supplement to this comment:

https://www.biogeosciences-discuss.net/bg-2018-148/bg-2018-148-AC2-supplement.pdf

**Supplement:**

Title: Effect of elevated $p$CO$_2$ on trace gas production during an ocean acidification mesocosm experiment Author(s): Sheng-Hui Zhang et al. MS No.: bg-2018-148

General Comments

The study examines production of volatile sulfur and halocarbon compounds in mesocosms of seawater with different dissolved carbon concentrations. The premise is to examine the impact of ocean acidification on gas production.

This is an okay idea. One major concern, however, is that the study was only five-weeks long, and there was no pretreatment of the phytoplankton. Thus, it is not really a global change test, but rather it is a test of acid shock on phytoplankton. I suppose this is interesting. Also, it appears to me that some of the data on temporal changes in chemistry and biology in the mesocosms have been published previously by Liu et al. (2017). Figure 1 is identical to Figure 1 and Figure 2 in Liu et al. (2107) and, at least, two panels in Figure 2 are in Figure 3 in Liu et al. (2017).

Thus, only the data in Figure 3 are new. Unfortunately, you cannot publish the same data twice. Elsevier, the publisher of Marine Environmental Research, owns the copyright to those figures.

Specific Comments

1) The abstract reads well.

2) The introduction is okay. However, it ends a bit abruptly. As written it is mostly a review of literature ending in an objective to do more research. Although a research objective is good, research should be question driven and present a testable, falsifiable hypothesis. In this case, what do you hope to learn in a 5-week study? (This seems short term to me.)

3) The methods seem appropriate, to me.

4) The results are okay. However, the discussion about the role of bacteria in DMSP dynamics, on page 10 and 11, seems like speculation to me. Where are the data on bacteria in the mesocosms? Speculation is okay, but data is better.

5) Than many correlations in the text could go in a table. This would make the text more readable.

6) Much of the discussion on page 13 is literature rather than interpretation. Rather than merely list other studies, compare results quantitatively. Did the other studies have CH$_3$I production rates that were similar to yours?

Technical Comments

1) Line 31 & 36: report the percentages as whole integers. It is nearly impossible to measure accurately to 0.1%.

2) Line 48: 'human activity' and 'anthropogenic' are the same. You do not need both in the sentence.

3) Line 69: delete the sentence 'several studies have already, etc.' in the following sentence, replace 'majority' with 'several studies have shown a negative impact, etc.'

4) Line 78: perhaps start a new paragraph with 'halocarbons'

5) Line 189 to 192: delete. This is not an appropriate topic sentence, and it is from the introduction. No need to repeat here.

6) Line 192: delete the sentence and put (Fig. 3) in the following sentence.

7) Line 209: round '29.2%' to the '29%'.

8) Line 228: why Yu et al., unpublished data? Why not include the data here?

9) Line 258: the sentence does not make sense. Do you mean 'attributed to biology' rather than 'involve'. Also delete the quotes around 'biogenic'. Why use quotes for an adjective?

Sorry but I cannot overlook the attempt to publish the same data in two papers. I realize that data from one paper can be used in another, but this needs to acknowledge the first paper and copywrite.

Response to Reviewer #2:

Dear Reviewer #2:

We are grateful to your review of this paper and would like to express our thanks for the helpful and constructive comments. We have revised the manuscript and addressed all the comments. The main changes we made are as follows:

General Comments

The study examines production of volatile sulfur and halocarbon compounds in mesocosms of seawater with different dissolved carbon concentrations. The premise is to examine the impact of ocean acidification on gas production.

This is an okay idea. One major concern, however, is that the study was only five-weeks long, and there was no pretreatment of the phytoplankton. Thus, it is not really a global change test, but rather it is a test of acid shock on phytoplankton. I suppose this is interesting.

Thanks for the reviewer's suggestion and we also agree with that the mesocosm experiment is a test of acid shock on phytoplankton. In addition, simple pretreatment was conducted before the mesocosm experiment as described in Huang et al. (2018). Briefly, before being introduced into the mesocosms, the three phytoplankton species and their associated bacteria were cultured in autoclaved, pre-filtered seawater from Wuyuan Bay at 16 ℃ (similar to the in situ temperature of Wuyuan Bay) without any addition of nutrients. Cultures were continuously aerated with filtered ambient air containing 400 μatm of $CO_2$ within plant chambers (HP1000G-D, Wuhan Ruihua Instrument & Equipment, China) at a constant bubbling rate of 300 ml min$^{-1}$. The culture medium was renewed every 24 h to maintain the cells of each phytoplankton species in exponential growth. We have added these pretreatment in the revised manuscript.

P6, L135-P7, L141 "Before being introduced into the mesocosms, the three phytoplankton species were cultured in autoclaved, pre-filtered seawater from Wuyuan Bay at 16 ℃ (similar to the in situ temperature of Wuyuan Bay) without any addition of nutrients. Cultures were continuously aerated with filtered ambient air containing 400 µatm of $CO_2$ within plant chambers (HP1000G-D, Wuhan Ruihua Instrument & Equipment, China) at a constant bubbling rate of 300 mL $min^{-1}$. The culture medium was renewed every 24 h to maintain the cells of each phytoplankton species in exponential growth."

Also, it appears to me that some of the data on temporal changes in chemistry and biology in the mesocosms have been published previously by Liu et al. (2017). Figure 1 is identical to Figure 1 and Figure 2 in Liu et al. (2017) and, at least, two panels in Figure 2 are in Figure 3 in Liu et al. (2017).

Thus, only the data in Figure 3 are new. Unfortunately, you cannot publish the same data twice. Elsevier, the publisher of Marine Environmental Research, owns the copyright to those figures.

We agree with the reviewer's suggestion. In the revised manuscript, we deleted the conflicting figures and only described these results simply.

P9, L187-L192"The phytoplankton growth process was divided into three phases in terms of variations in Chl *a* concentrations in the mesocosm experiments as described in Liu et al. (2017): i) the logarithmic growth phase (phase I, days 0–13), ii) a plateau phase (phase II, days 13–23, bloom period), and iii) a secondary plateau phase (phase III, days 23–33) attained after a decline in biomass from a maximum in phase II."

P9, L200-L204 "*Emiliania huxleyi* was only found in phase I and its maximal concentration reached 310 cells $mL^{-1}$ according to the results of microscopic inspection. *Thalassiosira weissflogii* was found throughout the entire period in each bag, but the maximum concentration was 8,120 cells $mL^{-1}$, which was far less than the concentration of *Phaeodactylum tricornutum* with a maximum density of about 1.5 million cells $mL^{-1}$ (Liu et al., 2017)."

Specific Comments

1) The abstract reads well.

Thanks for the reviewer's ratification.

2) The introduction is okay. However, it ends a bit abruptly. As written it is mostly a review of literature ending in an objective to do more research. Although a research objective is good, research should be question driven and present a testable, falsifiable hypothesis. In this case, what do you hope to learn in a 5-week study? (This seems short term to me.)

We agree with the reviewer's suggestion and have made modification in the revised manuscript.

P5, L97-103 "DMS and halocarbons play a significant role in the global climate and perhaps act a greater extent in the future. Meanwhile, the combined picture arising from existing studies is that the response of communities to OA is not predictable and further studies were required. Based on the controversial results about OA on DMS and halocarbons production, a mesocosm experiment was conducted in Wu Yuan Bay, Xiamen. The aim of this study was to investigate the influence of elevated $p$CO$_2$ on diatoms and coccolithophores and to further understand how the productions of DMS and halocarbons respond to OA."

During this experiment, the nutrient concentrations (dissolved inorganic nitrogen (DIN) and phosphate) in phase II were below or close to the detection limit, though the Chl $a$ concentration still maintained a relatively high concentration after 5 weeks incubation. We think that the stored nutrients in diatom cells might contribute to the biomass increase even after the depletion of nutrients in the surrounding seawater (Goldman et al., 1979; Sommer, 1989). Meanwhile, DMS, DCB and CH$_3$I concentration decreased significantly after 5 weeks incubation. Therefore, 5 weeks incubation is appropriate to this experiment.

3) The methods seem appropriate, to me.
Thanks for the reviewer's ratification.

4) The results are okay. However, the discussion about the role of bacteria in DMSP dynamics, on page 10 and 11, seems like speculation to me. Where are the data on bacteria in the mesocosms? Speculation is okay, but data is better.
We agree with the reviewer's suggestion. We have added the DCB data in the revised manuscript.

P8, L167-L175 "2.5 *Enumeration of DMSP-consuming bacteria (DCB)*

[revised manuscript text omitted]

6) Much of the discussion on page 13 is literature rather than interpretation. Rather than merely list other studies, compare results quantitatively. Did the other studies have $CH_3I$ production rates that were similar to yours?

We agree with reviewer's suggestion and have made the modification in the revised manuscript.

P13, L280-288 "The temporal dynamics of $CH_3I$ in the HC and LC treatments are shown in Fig. 3-D. The $CH_3I$ concentrations in the LC treatment varied from 0.38 to 12.61 pmol $L^{-1}$, with a mean of 4.76 pmol $L^{-1}$. The $CH_3I$ concentrations in the HC treatment ranged between 0.44 and 8.78 pmol $L^{-1}$, with a mean of 2.88 pmol $L^{-1}$. The maximum $CH_3I$ concentrations in the HC and LC treatments were both observed on day 23. The range of $CH_3I$ concentrations during this experiment was similar to that measured in the mesocosm experiment ($< 1{\sim}10$ pmol $L^{-1}$) in Kongsfjorden conducted by Hopkins et al. (2013). In addition, the mean $CH_3I$ concentration in the LC treatment was similar to that measured in the East China Sea, with an average of 5.34 pmol $L^{-1}$ in winter and 5.74 pmol $L^{-1}$ in summer (Yuan et al., 2015)."

Technical Comments

1) Line 31 & 36: report the percentages as whole integers. It is nearly impossible to measure accurately to 0.1%.

Thanks for the reviewer's suggestion. We have modified this in the revised manuscript.

P2, L36-43 "During the logarithmic growth phase (phase I), DMS concentrations in high $pCO_2$ mesocosms (HC, 1000 μatm) were 28% lower than those in low $pCO_2$ mesocosms (LC, 400 μatm). Elevated pCO2 led to a delay in DCB concentrations attached to Thalassiosira weissflogii and Phaeodactylum tricornutum and finally resulted in the delay of DMS concentration in the HC treatment. Unlike DMS, the elevated $pCO_2$ did not affect DMSP production ability of Thalassiosira weissflogii or Phaeodactylum tricornuntum throughout the 5 weeks culture. A positive relationship was detected between CH3I and Thalassiosira weissflogii and Phaeodactylum tricornuntum during the experiment, and there was a 40% reduction in mean $CH_3I$ concentrations in the HC mesocosms."

2) Line 48: 'human activity' and 'anthropogenic' are the same. You do not need both in the sentence.

We agree with the reviewer's suggestion and have modified this in the revised manuscript.

P3, L53-56 "Anthropogenic emissions have increased the fugacity of atmospheric carbon dioxide ($pCO_2$) from the pre-industrial value of 280 μatm to the present-day value of over 400 μatm, and these values will further increase to 800–1000 μatm by the end of this century according to the Intergovernmental Panel on Climate Change (Gattuso et al., 2015)."

3) Line 69: delete the sentence 'several studies have already, etc.' in the following sentence, replace 'majority' with 'several studies have shown a negative impact, etc.'

We agree with the reviewer's suggestion and have made modification in the revised manuscript.

P4, L77-80 "Several studies have shown a negative impact of decreasing pH on DMS-production capability (Hopkins et al., 2010; Avgoustidi et al., 2012; Archer et al., 2013; Webb et al., 2016), while others have found either no effect or a positive effect (Vogt et al., 2008; Hopkins and Archer, 2014)."

4) Line 78: perhaps start a new paragraph with 'halocarbons'

Thanks for the reviewer's suggestion. We have started a new paragraph with 'halocarbons' in the revised manuscript..

5) Line 189 to 192: delete. This is not an appropriate topic sentence, and it is from the introduction. No need to repeat here.

Thanks for the reviewer's suggestion. We have deleted this sentence in the revised manuscript.

6) Line 192: delete the sentence and put (Fig. 3) in the following sentence.

We agree with the reviewer's suggestion and have made the modification in the revised manuscript.

P9, L207-P10, L209 "At the beginning of the experiment, the mean DMS, DMSP and DCB concentrations were all low in both treatments due to the low concentrations of DMS, DMSP and DCB in the original fjord water and possible loss during the filtration procedure (Fig. 2)."

7) Line 209: round '29.2%' to the '29%'.

We agree with the reviewer's suggestion and have made the modification in the revised manuscript.

P11, L231-234 "However, a significant 29% reduction in DMS concentrations was detected in the HC treatment compared with the LC treatment ($p$ = 0.016), though no statistical difference for DCB concentrations was found between the LC and HC treatments during phase I."

8) Line 228: why Yu et al., unpublished data? Why not include the data here?

We agree with the reviewer's suggestion. We have added the DCB data in the revised manuscript.

P8, L167-L175 "2.5 *Enumeration of DMSP-consuming bacteria (DCB)*

The number of DMSP-consuming bacteria (DCB) was estimated using the most probable number (MPN) methodology. The MPN medium consisted of a mixture (1:1 v/v) of sterile artificial sea water (ASW) and mineral medium (Visscher et al., 1991), 3 mL of which was dispensed in 6 mL test tubes, which were closed off by an over-sized cap, allowing gas exchange. Triplicate dilution series were set up. All test tubes contained 1 mmol L$^{-1}$ DMSP as the sole organic carbon source and were kept at 30 °C in the dark. After 2 weeks, the presence/absence of bacteria in the tubes was verified by DAPI staining (Porter and Feig, 1980). Three tubes containing 3 mL ASW without substrate were used as controls."

P10, L217-L224 "Compared with DMSP, DMS and DCB concentrations showed similar trends during the mesocosm experiment. DMS concentrations in the LC and HC treatments were 1.03 and 0.74 nmol L$^{-1}$, respectively, while DCB concentrations in the LC and HC treatments were $0.20 \times 10^6$ and $0.16 \times 10^6$ cells mL$^{-1}$. DMS and DCB concentrations did not increase significantly during phase I, but began to increase rapidly on day 15. DCB concentrations in the LC and HC treatments peaked on days 21 ($11.65 \times 10^6$ cells mL$^{-1}$) and 23 ($10.70 \times 10^6$ cells mL$^{-1}$), while DMS concentrations in the LC and HC treatments peaked on days 25 (112.1 nmol L$^{-1}$) and 30 (101.9 nmol L$^{-1}$). Both DMS and DCB concentrations began to decrease obviously during phase III."

P11, L231-L234 "However, a significant 29% reduction in DMS concentrations was detected in the HC treatment compared with the LC treatment ($p = 0.016$), though no statistical difference for DCB concentrations was found between the LC and HC treatments during phase I."

P11, L244-L246 "In addition, a significant positive relationship was also observed between DMS and DCB (r = 0.643, $p < 0.01$ in the LC treatment; r = 0.544, $p < 0.01$ in the HC treatment) during this experiment."

P12, L250-L253 "Moreover, DCB peaked on days 21 ($11.65 \times 10^6$ cells mL$^{-1}$) and 23 ($10.70 \times 10^6$ cells mL$^{-1}$) in the LC and HC treatments, respectively, as shown in Fig. 2-C. Similar to DMS, DCB was also delayed in the HC mesocosm compared to that in the LC mesocosm."

9) Line 258: the sentence does not make sense. Do you mean 'attributed to biology' rather than 'involve'. Also delete the quotes around 'biogenic'. Why use quotes for an adjective?

We agree with the reviewer's suggestion and have deleted this sentence in the revised manuscript.

[revised manuscript text omitted]
* | 0.635** | 0.954** | 0.803** | 0.143 | -0.257 | 0.267 | 0.834** | 0.559 | 0.820** | 1 |

---

## Author Response (AR2)

**An itemized response (blue words) to the associate editor's comments and suggestions**

We have carefully considered the associate editor 's comments and suggestions and conducted the revision seriously. Besides, our manuscript has been carefully edited by a native English speaker, as indicated in the following certificate of language editing. We are very thankful to the associate editor for all the valuable comments and helpful suggestions to improve this manuscript.

Associate Editor Decision: Publish subject to minor revisions (review by editor) (20 Sep 2018) by Tina Treude

Comments to the Author:

Dear Dr. Zhuang and Co-Workers,

Let me start by apologizing for the delay. I decided to review your revised manuscript by myself, since the former referees where not available anymore, but then I got interrupted by a three-week travel through conferences in Europe.

I read through the referee's comments, your responses, and the revised manuscript and most of the scientific recommendations have been implemented. There are some points left that I will further explain below. There are a couple, but they should be easy for you to implement.

The main issue that remains is the poor scientific English of the manuscript. I am not a native English speaker myself, but I fear your manuscript needs major copy-editing. Biogeosciences does copy-editing but only to a certain degree. Do you have a native English speaker in your institution who could revise the manuscript for you to improve its readability? I will try my best to give you some advice in my detailed comments below, but it is a major undertaking and beyond my duties.

With kind regards

Tina Treude

Thanks for the associate editor's suggestion and our manuscript has been carefully edited by a native English speaker.

[Figure]

EditorBar Language Editing
No. 35, Tsinghua East Road, Beijing, China 100083
Email: runse@editorbar.com Phone: +86-10-5620-8614

**CERTIFICATE OF LANGUAGE EDITING**

The English writing of the following manuscript was carefully edited by a native English speaker.

**Manuscript Information**

| | |
|---|---|
| ID | AE201801070068-R2 |
| Editing date | 2018-09-30 |
| Title | Effect of elevated pCO2 on trace gas production during an ocean acidification mesocosm experiment |
| Corresponding author | Gui-peng Yang |
| Language writing before editing | □ Very poor    □ Poor    ■ Fair    □ Good    □ Very good    □ Excellent |
| Recommendation after language editing | □ Submitting to target journal directly
■ Submitting to target journal after minor revision
□ Re-editing required after major revision
□ Not suitable for publication |
| Overview comments | |

**Edited by**

**William K.**
Ph. D
North Carolina State University
Language Editing

**Certificate Issued by**

**Dr. Jason Qee**

*Jason qee*

Editor in Chief
Editorbar Language Editing, Beijing, China
runse@editorbar.com   www.editorbar.com

Certificate link: www.editorbar.com/order/cert/AE201801070068-R2

1. General Comments

- The manuscript has too many abbreviations, which makes it hard to read. I strongly recommend reducing the abbreviations to a minimum. For example: write 'ocean acidification' not 'OA', write 'high $p$CO$_2$' and 'low $p$CO$_2$' not 'HC' and 'LC', write 'iodine' not 'I'. There are many more

unnecessary abbreviations (DCB, MPN, ASW etc.). Please do not abbreviate if a term is used less than 5 times or if the term consists of only two words.

Thanks for the associate editor's suggestion. We have modified the abbreviations according to the associate editor's suggestion in the revised manuscript.

- Species names are written out (Phaeodactylum tricornuntum) the first time they are mentioned, after that the genus name is abbreviated (P. tricornuntum)

Thanks for the associate editor's suggestion. We have solved this problem in the revised manuscript.

- Figure and table captions must be self-explanatory. Please provide definition of all abbreviations used in the figure and table captions and figures and tables.

Thanks for the associate editor's suggestion. We have provided definition of all abbreviations used in the figure and table captions and figures and tables in the revised manuscript.

2. Scientific Comments

L33: Add species names in brackets after "phytoplankton species"

Thanks for the associate editor's suggestion. We have added species names after "phytoplankton species"

L32-36 "A mesocosm experiment was conducted in Wuyuan Bay (Xiamen), China to investigate the effects of elevated $p$CO$_2$ on the phytoplankton species *Phaeodactylum tricornutum* (*P. tricornutum*), *Thalassiosira weissflogii* (*T. weissflogii*) and *Emiliania huxleyi* (*E. huxleyi*) and their production ability of dimethylsulfide (DMS), dimethylsulfoniopropionate (DMSP), as well as four halocarbon compounds bromodichloromethane (CHBrCl$_2$), methyl bromide (CH$_3$Br), dibromomethane (CH$_2$Br$_2$) and iodomethane (CH$_3$I)."

L102: This paragraph should end with a hypothesis. "respond to OA" is too vague.

Thanks for the associate editor's suggestion. We have reworded this section in the revised manuscript.

L97-105 "Taken together, the data indicate that the response of DMS and halocarbon release to elevated $p$CO$_2$ is complex and controversial. DMS and halocarbons play a significant role in the global climate and will perhaps act to a greater extent in the future. An intermediate step between laboratory and natural community field experiments was designed in this study to understand the response of the release of DMS and halocarbon to ocean acidification in Chinese coastal seas using isolates of non-axenic phytoplankton added to filtered natural water. We hypothesized that the response of DMS and halocarbon release to elevated $p$CO$_2$ in natural seawater can be better presented after minimizing the shifting composition of the natural phytoplankton and microbial communities. "

L104: "General experimental device" What do you mean here? The setup? Device sounds strange.

Thanks for the associate editor's suggestion. We have reworded this title in the revised manuscript.

L107 "*2.1 Experimental setup*"

L105: Start this paragraph with the main purpose of the experiment

Thanks for the associate editor's suggestion. We have started this paragraph with the main purpose of the experiment in the revised manuscript.

L108-111 "To investigate the response of DMS and halocarbon release to ocean acidification, a mesocosm experiment was carried out on a floating platform (set in seawater, about 150 m from the shore) at the Facility for Ocean Acidification Impacts Study of Xiamen University (FOANIC-XMU, 24.52°N, 117.18°E) (for full technical details of the mesocosms, see Liu et al. 2017)."

L105: Does "floating platform" mean in the water? Offshore? Provide some details.

The "floating platform" was set in the water and we have provided some details in the revised manuscript.

L108-111 "To investigate the response of DMS and halocarbon release to ocean acidification, a mesocosm experiment was carried out on a floating platform (set in seawater, about 150 m from the shore) at the Facility for Ocean Acidification Impacts Study of Xiamen University (FOANIC-XMU, 24.52°N, 117.18°E) (for full technical details of the mesocosms, see Liu et al. 2017)."

L114: Define low and high $p$CO$_2$ levels.

Thanks for the associate editor's suggestion. We have defined low and high $p$CO$_2$ levels in the revised manuscript.

L118-120 "To set the low (400 μatm) and high $p$CO$_2$ (1000 μatm) levels, we added Na$_2$CO$_3$ solution and CO$_2$ saturated seawater to the mesocosm bags to alter total alkalinity and dissolved inorganic carbon (Gattuso et al., 2010; Riebesell et al., 2013)."

L119-122: I am not sure I am able to follow what is meant by coastal environment (algae) and just environment (trace gases). Is there a difference between the environments? How large was the filter (pore size and type) to remove the algae and bacteria? Which trace gases do you mean? How did you measure them?

Thanks for the associate editor's comment. An ultrafiltration water purifier (0.01 μm, MU801-4T, Midea, Guangdong, China) was used to remove the algae and bacteria in the natural seawater before this experiment. The trace gases in this manuscript referred to DMS, CHBrCl$_2$, CH$_3$Br, CH$_2$Br$_2$ and CH$_3$I, and their determination method section 2.3.

L140-141: "no meaningful numbers" is not a scientific term. Please provide a minimum threshold, such as "less than xxx cells per Liter".

Thanks for the associate editor's suggestion. We have modified this section in the revised manuscript.

L125-127 "Bacterial abundance in the pre-filtered water was less than $10^3$ cell mL$^{-1}$, which was three magnitudes lower than the bacterial abundance in the natural water and close to the detection limit of the flow cytometer."

L166-174: As requested by the referees you added the enumeration of the DMSP-consuming bacteria. However, unless I missed it, the data are nowhere presented or discussed in the text and Fig. S1 is not cited. Please add accordingly. Please integrate Fig. S1 into the main manuscript. It should not be a supplementary, since the data are essential to this study.

Thanks for the associate editor's suggestion. We have modified this section in the revised manuscript.

"Similar to DMS, DMSP-consuming bacteria also maintained a low level during phase I (mean of $0.57 \times 10^6$ and $0.40 \times 10^6$ cells mL$^{-1}$ in the low $p$CO$_2$ and high $p$CO$_2$ treatments, respectively). DMSP-consuming bacterial concentrations respectively peaked on days 19 ($11.65 \times 10^6$ cells mL$^{-1}$) and 21 ($10.70 \times 10^6$ cells mL$^{-1}$) in the low $p$CO$_2$ and high $p$CO$_2$ treatments."

"significant reductions in mean DMS concentration (28%) and DMSP-consuming bacteria (29%) were detected during phase I in the high $p$CO$_2$ treatment compared with those in the low $p$CO$_2$ treatment, indicating that elevated $p$CO$_2$ inhibits DMSP-consuming bacteria and DMS production during the logarithmic growth phase."

[Figure]

**Fig. 2** Temporal changes in dimethylsulfide (DMS), dimethylsulfoniopropionate (DMSP), DMSP-consuming bacteria concentrations in the high $pCO_2$ (1,000 μatm, black squares) and low $pCO_2$ (400 μatm, white squares) mesocosms. Data are mean ± standard deviation, n = 3 (triplicate independent mesocosm bags) (Origin 8.0). ''

L184: Provide a method to "well control" temperature and salinity.

The mesocosm bags were set in seawater and the temperature was controlled by the surrounding seawater. In addition, these mesocosm bags were closed and this experiment was completed during winter. Therefore, the effect of evaporation on salinity was not obvious and salinity maintained at 29 during this study.

L201: I could not find a method for "microscopic inspections". Please add.

Thanks for the associate editor's suggestion. The *P. tricornuntum*, *T. weissflogii* and *E. huxleyi* data were determined by Liu N and Gao K, and we have removed out these figures from this manuscript according to the reviewer's suggestion. Therefore, we think that it is appropriate to cite

a reference in this section.

L213-214 "*E. huxleyi* was only found in phase I and its maximal concentration reached 310 cells mL$^{-1}$ according to the results of Liu et al. (2017). "

L300-301: Please shortly elaborate what the major differences between the cited literature are, because you can't expect the reader to first read all the literature to understand your argument.

Thanks for the associate editor's suggestion. We have modified this section in the revised manuscript.

L306-309 "This result is in accordance with Hopkins et al. (2010) and Webb et al. (2015) who also reported that elevated $p$CO$_2$ leads to a reduction in iodocarbon concentrations, but in contrast to the findings of Hopkins et al. (2013) and Webb et al. (2016) who showed that elevated $p$CO$_2$ does not significantly affect the iodocarbon concentrations in the mesocosms."

L301-302: Which mesocosm experiment? All of them? 40.2% reduction of what? I also think the reviewers asked you to round before the digit (40%).

Thanks for the associate editor's suggestion. We have modified this section in the revised manuscript.

L304-306 "Furthermore, the mean CH$_3$I concentration measured in the high $p$CO$_2$ treatment was significantly lower (40%) than that measured in the low $p$CO$_2$ treatment during the mesocosm experiment."

L313: I think the reviewers asked you to round before the digit (28%).

Thanks for the associate editor's suggestion. We have modified this section in the revised manuscript.

L321-323 "During the logarithmic growth phase, the elevated $p$CO$_2$ led to a reduction in mean

DMSP-consuming bacteria (29%) and DMS concentration (28%) compared with those in the low

$p$CO$_2$ treatment."

L316-317: I still do not fully understand the part about the trace gases (see my earlier comment).

Maybe it just needs clarification.

Thanks for the associate editor's suggestion. We have modified this section in the revised

manuscript.

L114-130 "Filtered (0.01 μm ultrafiltration water purifier, MU801-4T, Midea, Guangdong, China)

in situ seawater was pumped into the six bags simultaneously within 24 h. A known amount of

NaCl solution was added to each bag to calculate the exact volume of seawater in the bags,

according to a comparison of the salinity before and after adding salt (Czerny et al., 2013). The

initial in situ $p$CO$_2$ was about 650 μatm. To set the low (400 μatm) and high $p$CO$_2$ (1000 μatm)

levels, we added Na$_2$CO$_3$ solution and CO$_2$ saturated seawater to the mesocosm bags to alter total

alkalinity and dissolved inorganic carbon (Gattuso et al., 2010; Riebesell et al., 2013).

Subsequently, during the whole experimental process, air at the ambient (400 μatm) and elevated

$p$CO$_2$ (1000 μatm) concentrations was continuously bubbled into the mesocosm bags using a CO$_2$

Enricher (CE-100B, Wuhan Ruihua Instrument & Equipment Ltd., Wuhan, China). Seawater taken

from the coastal environment was first filtered to remove algae and their attached bacteria before

usage in mesocosm bags. Bacterial abundance in the pre-filtered water was less than $10^3$ cell mL$^{-1}$,

which was three magnitudes lower than the bacterial abundance in the natural water and close to

the detection limit of the flow cytometer. The trace gases, including DMS, bromodichloromethane

(CHBrCl$_2$), methyl bromide (CH$_3$Br), dibromomethane (CH$_2$Br$_2$), and iodomethane (CH$_3$I)

produced in the environment did not affect the mesocosm trace gas concentrations after the bags were sealed."

L315-317: Is this really a main conclusion or just a technical detail? I don't understand why it is mentioned in this chapter.

Thanks for the associate editor's suggestion. We have deleted this part in the revised manuscript.

L321: Details needed for "a range of biological parameters"

Thanks for the associate editor's suggestion. We have modified this section in the revised manuscript.

L326-327 "Affected by the filtration procedure, three bromocarbons compounds measured in this study were not correlated with *P. tricornuntum* and *T. weissflogii*, and Chl *a*."

L323-324: "biological parameters" is too vague and needs detail. What do you mean by "biological control"? Specify

According to the associate editor's suggestion, we have modified this section in the revised manuscript.

L328-331 "The temporal dynamics of $CH_3I$, combined with strong correlations with *P. tricornuntum* and *T. weissflogii*, and Chl *a*, indicate that *P. tricornuntum* and *T. weissflogii* play a critical role controlling $CH_3I$ concentrations."

3. Scientific English and structure

L86: change "that they may account" to "accounting"

Thanks for the associate editor's suggestion. We have modified this part in the revised manuscript.

L85-88 "Halocarbons also play a significant role in the global climate because they are linked to tropospheric and stratospheric ozone depletion and a synergistic effect of chlorine and bromine

species has been reported accounting for approximately 20% of the polar stratospheric ozone depletion (Roy et al., 2011)."

L93: Through = in

Thanks for the associate editor's suggestion. We have modified this part in the revised manuscript.

L92-94 "Hopkins et al. (2010) and Webb et al. (2015) measured lower concentrations of several iodocarbons, while bromocarbons were unaffected by elevated $p$CO$_2$ in two acidification experiments."

L93: Write: "in addition, another mesocosm..."

Thanks for the associate editor's suggestion. We have modified this part in the revised manuscript.

L94-96 "In addition, another mesocosm study did not elicit significant differences from any halocarbon compounds at up to 1,400 µatm $p$CO$_2$ (Hopkins et al., 2013)."

L96-102: This paragraph is poorly written and hard to understand. Please revise thoroughly and let it be edited by a native English speaker.

Thanks for the associate editor's suggestion. This section as well as the whole manuscript was reworded and edited by a native English speaker.

L97-105 "Taken together, the data indicate that the response of DMS and halocarbon release to elevated $p$CO$_2$ is complex and controversial. DMS and halocarbons play a significant role in the global climate and will perhaps act to a greater extent in the future. An intermediate step between laboratory and natural community field experiments was designed in this study to understand the response of the release of DMS and halocarbon to ocean acidification in Chinese coastal seas using isolates of non-axenic phytoplankton added to filtered natural water. We hypothesized that the response of DMS and halocarbon release to elevated $p$CO$_2$ in natural seawater can be better

presented after minimizing the shifting composition of the natural phytoplankton and microbial communities. **"**

L97 (and many other places throughout the manuscript): Please do not use "Meanwhile,". It is a temporal term and is consistently misused in this manuscript.

Thanks for the associate editor's suggestion. This section as well as the whole manuscript was reworded and edited by a native English speaker.

L97-105 "Taken together, the data indicate that the response of DMS and halocarbon release to elevated $p$CO$_2$ is complex and controversial. DMS and halocarbons play a significant role in the global climate and will perhaps act to a greater extent in the future. An intermediate step between laboratory and natural community field experiments was designed in this study to understand the response of the release of DMS and halocarbon to ocean acidification in Chinese coastal seas using isolates of non-axenic phytoplankton added to filtered natural water. We hypothesized that the response of DMS and halocarbon release to elevated $p$CO$_2$ in natural seawater can be better presented after minimizing the shifting composition of the natural phytoplankton and microbial communities. "

Line 98: I do not understand why you write "were required" (past tense)

Thanks for the associate editor's suggestion. This section as well as the whole manuscript was reworded and edited by a native English speaker.

L97-105 "Taken together, the data indicate that the response of DMS and halocarbon release to elevated $p$CO$_2$ is complex and controversial. DMS and halocarbons play a significant role in the global climate and will perhaps act to a greater extent in the future. An intermediate step between laboratory and natural community field experiments was designed in this study to understand the

response of the release of DMS and halocarbon to ocean acidification in Chinese coastal seas using isolates of non-axenic phytoplankton added to filtered natural water. We hypothesized that the response of DMS and halocarbon release to elevated $p$CO$_2$ in natural seawater can be better presented after minimizing the shifting composition of the natural phytoplankton and microbial communities. "

L99-100: "a mesocosm experiment was conducted" Are you talking about your study? Unclear.

Thanks for the associate editor's suggestion. This section as well as the whole manuscript was reworded and edited by a native English speaker.

L97-105 "Taken together, the data indicate that the response of DMS and halocarbon release to elevated $p$CO$_2$ is complex and controversial. DMS and halocarbons play a significant role in the global climate and will perhaps act to a greater extent in the future. An intermediate step between laboratory and natural community field experiments was designed in this study to understand the response of the release of DMS and halocarbon to ocean acidification in Chinese coastal seas using isolates of non-axenic phytoplankton added to filtered natural water. We hypothesized that the response of DMS and halocarbon release to elevated $p$CO$_2$ in natural seawater can be better presented after minimizing the shifting composition of the natural phytoplankton and microbial communities. "

L100: Write "The aim of the present study..."

Thanks for the associate editor's suggestion. This section as well as the whole manuscript was reworded and edited by a native English speaker.

L97-105 "Taken together, the data indicate that the response of DMS and halocarbon release to elevated $p$CO$_2$ is complex and controversial. DMS and halocarbons play a significant role in the

global climate and will perhaps act to a greater extent in the future. An intermediate step between laboratory and natural community field experiments was designed in this study to understand the response of the release of DMS and halocarbon to ocean acidification in Chinese coastal seas using isolates of non-axenic phytoplankton added to filtered natural water. We hypothesized that the response of DMS and halocarbon release to elevated $p$CO$_2$ in natural seawater can be better presented after minimizing the shifting composition of the natural phytoplankton and microbial communities. "

L101: Delete "further", delete "s" in "productions"

Thanks for the associate editor's suggestion. This section as well as the whole manuscript was reworded and edited by a native English speaker.

L97-105 "Taken together, the data indicate that the response of DMS and halocarbon release to elevated $p$CO$_2$ is complex and controversial. DMS and halocarbons play a significant role in the global climate and will perhaps act to a greater extent in the future. An intermediate step between laboratory and natural community field experiments was designed in this study to understand the response of the release of DMS and halocarbon to ocean acidification in Chinese coastal seas using isolates of non-axenic phytoplankton added to filtered natural water. We hypothesized that the response of DMS and halocarbon release to elevated $p$CO$_2$ in natural seawater can be better presented after minimizing the shifting composition of the natural phytoplankton and microbial communities. "

L119-122: Change to: "Seawater taken from the coastal environment was first filtered to remove algae and their attached bacteria before usage in mesocosms. Trace gases produced..."

Thanks for the associate editor's suggestion. We have modified this part in the revised manuscript.

L123-130 "Seawater taken from the coastal environment was first filtered to remove algae and their attached bacteria before usage in mesocosm bags. Bacterial abundance in the pre-filtered water was less than $10^3$ cell $mL^{-1}$, which was three magnitudes lower than the bacterial abundance in the natural water and close to the detection limit of the flow cytometer. The trace gases, including DMS, bromodichloromethane ($CHBrCl_2$), methyl bromide ($CH_3Br$), dibromomethane ($CH_2Br_2$), and iodomethane ($CH_3I$) produced in the environment did not affect the mesocosm trace gas concentrations after the bags were sealed."

L134-141: This entire paragraph should be moved up to the beginning of the "Algal strains" chapter, because this step happens prior to the mesocosm experiments.

According to the associate editor's suggestion, we have modified this section in the revised manuscript.

L132-148 "Before being introduced into the mesocosms, the three phytoplankton species *Phaeodactylum tricornutum* (*P. tricornutum*), *Thalassiosira weissflogii* (*T. weissflogii*) and *Emiliania huxleyi* (*E. huxleyi*) were cultured in autoclaved, pre-filtered seawater from Wuyuan Bay at 16°C (similar to the in situ temperature of Wuyuan Bay) without any addition of nutrients. Cultures were continuously aerated with filtered ambient air containing 400 µatm of $CO_2$ within plant chambers (HP1000G-D, Wuhan Ruihua Instrument & Equipment, China) at a constant bubbling rate of 300 mL $min^{-1}$. The culture medium was renewed every 24 hrs to maintain the cells of each phytoplankton species in exponential growth. When the experiment began, these three phytoplankton species were inoculated into the mesocosm bags, with an initial diatom/coccolithophorid cell ratio of 1:1. The initial concentrations of *P. tricornuntum*, *T. weissflogii*, and *E. huxleyi* inoculated into the mesocosm were 10, 10, and 20 cells $mL^{-1}$,

respectively. *P. tricornuntum* and *T. weissflogii* were obtained from the Center for Collections of Marine Bacteria and Phytoplankton of the State Key Laboratory of Marine Environmental Science (Xiamen University). *P. tricornuntum* was originally isolated from the South China Sea in 2004 and *T. weissflogii* was isolated from Daya Bay in the coastal South China Sea. *E. huxleyi* was originally isolated in 1992 from the field station of the University of Bergen (Raunefjorden; 60°18'N, 05°15'E). "

L143: delete "s" in hydrocarbons", delete "generally", replace "obtained" by "taken", add "the" after "from"

According to the associate editor's suggestion, we have modified this section in the revised manuscript.

L150-152 "DMS(P) and halocarbon samples were taken from the above mentioned mesocosm bags at 9 a.m., then all collected samples were transported into a dark cool box back to the laboratory onshore for analysis within 1 hrs."

L151: "h" = "hrs"

According to the associate editor's suggestion, we have modified this section in the revised manuscript.

L150-152 "DMS(P) and halocarbon samples were taken from the above mentioned mesocosm bags at 9 a.m., then all collected samples were transported into a dark cool box back to the laboratory onshore for analysis within 1 hr."

L167: Which experiment do you refer to in this sentence? Please add information (in which sample did you count the DCBs?).

According to the associate editor's suggestion, we have modified this section in the revised

manuscript.

L174-175 "The number of DMSP-consuming bacteria in the mesocosms was estimated using the most probable number methodology."

L169: Write "into 6 mL"

According to the associate editor's suggestion, we have modified this section in the revised manuscript.

L175-177 "The medium consisted of a mixture (1:1 v/v) of sterile artificial sea water and mineral medium (Visscher et al., 1991), 3 mL of which was dispensed into 6 mL test tubes, which were closed by an over-sized cap, allowing gas exchange."

L170: Delete "off"

According to the associate editor's suggestion, we have modified this section in the revised manuscript.

L175-177 "The medium consisted of a mixture (1:1 v/v) of sterile artificial sea water and mineral medium (Visscher et al., 1991), 3 mL of which was dispensed into 6 mL test tubes, which were closed by an over-sized cap, allowing gas exchange."

L181-204: This section is missing a discussion. You are just presenting results, but this is a combined "Results and Discussion" Chapter. Please add a discussion of the data and make literature comparisons where appropriate.

According to the associate editor's suggestion, we have added a discussion about this section in the revised manuscript.

L203-205 "$SiO_3^{2-}$ was undetectable during the entire experimental period, and was unlikely to be a limiting factor for phytoplankton growth during the experiment."

L210-213 "It is possible that *P. tricornutum* outcompeted *T. weissflogii* because of its higher surface to volume ratio and/or species-specific physiology, which would enhance the efficiency of nutrient uptake and related metabolism (Alessandrade et al., 2007)."

L347-348 "Alessandrade, M., Agnès, M., Shi, J., Pan, K., Chris, B.: Genetic and phenotypic characterization of *Phaeodactylum tricornutum* (Bacillariophyceae) accessions. J. Phycol., 43, 992–1009, 2007."

L215-218 "Previous studies have reported that the maximum specific growth rate of *T. weissflogii* and *P. tricornutum* is about 1.2 $d^{-1}$ (Li et al., 2014; Sugie and Yoshimura, 2016), while that of *E. huxleyi* is about 0.8 $d^{-1}$ (Xing et al., 2015). This might be the main reason why diatoms overwhelmingly outcompeted the coccolithophores during this experiment."

L408-409 "Li, Y. H., Xu, J. T., Gao, K.: Light-modulated responses of growth and photosynthetic performance to ocean acidification in the model diatom Phaeodactylum tricornutum. PLoS One 9, e96173, 2014."

L454-455 "Sugie, K., Yoshimura, T.: Effects of high $CO_2$ levels on the ecophysiology of the diatom *Thalassiosira weissflogii* differ depending on the iron nutritional status. ICES J. Mar. Sci. 73, 680–692, 2016."

L477-478 "Xing, T., Gao, K., Beardall, J.: Response of growth and photosynthesis of *Emiliania huxleyi* to visible and UV irradiances under different light regimes. Photochem. Photobiol. 91, 343–349, 2015."

L183: What does "well combined" mean? Not proper English.

According to the associate editor's suggestion, we have modified this section in the revised manuscript.

L190-191 "During the experiment, the seawater in each mesocosm was well mixed, and the temperature and salinity remained stable, with means of 16°C and 29, respectively, in all mesocosm bags."

L184: Move ",respetively" after "29"

According to the associate editor's suggestion, we have modified this section in the revised manuscript.

L190-191 "During the experiment, the seawater in each mesocosm was well mixed, and the temperature and salinity remained stable, with means of 16°C and 29, respectively, in all mesocosm bags."

L193-194: add ", respectively," after "41 µmol L-1" and "38 µmol L-1".

According to the associate editor's suggestion, we have modified this section in the revised manuscript.

L198-201 "The initial mean dissolved nitrate (including $NO_3^-$ and $NO_2^-$), $NH_4^+$, $PO_4^{3-}$ and silicate ($SiO_3^{2-}$) concentrations were 54, 20, 2.6 and 41 µmol $L^{-1}$, respectively for the low $p$CO$_2$ treatment and 52, 21, 2.4 and 38 µmol $L^{-1}$, respectively for the high $p$CO$_2$ treatment."

L206-215: The paragraphs 206-211 and 211-215 are a bit repetitive. They should be better combined.

According to the associate editor's suggestion, we have modified this section in the revised manuscript.

L220-229 "DMSP concentrations in the high $p$CO$_2$ and low $p$CO$_2$ treatments increased significantly along with the increase in Chl $a$ concentrations and algal cells, and remained relatively constant over the following days. A significant positive relationship was observed between DMSP and phytoplankton in the experiment (r = 0.961, $p$ < 0.01 for $P. tricornuntum$, r = 0.617, $p$ < 0.01 for $T. weissflogii$ in the low $p$CO$_2$ treatment, Table 2; r = 0.954, $p$ < 0.01 for $P. tricornuntum$, r = 0.743, $p$ < 0.01 for $T. weissflogii$ in the high $p$CO$_2$ treatment, Table 3). DMS was

maintained at a low level during phase I (mean of 1.03 nmol $L^{-1}$ in the low $pCO_2$ and 0.74 nmol $L^{-1}$ in the high $pCO_2$ treatments, respectively) compared with DMSP. DMS concentrations began to increase rapidly on day 15, peaked on day 25 in the low $pCO_2$ treatment (112.1 nmol $L^{-1}$) and on day 29 in the high $pCO_2$ treatment (101.9 nmol $L^{-1}$) respectively, and then decreased in the following days."

L223: delete "obviously"

According to the associate editor's suggestion, we have modified this section in the revised manuscript.

L227-229 "DMS concentrations began to increase rapidly on day 15, peaked on day 25 in the low $pCO_2$ treatment (112.1 nmol $L^{-1}$) and on day 29 in the high $pCO_2$ treatment (101.9 nmol $L^{-1}$) respectively, and then decreased in the following days."

L228 & 230: "In this study" and "throughout this study" is repetitive.

According to the associate editor's suggestion, we have modified this section in the revised manuscript.

L237-239 "In this study, no difference in mean DMSP concentrations was observed between the two treatments, indicating that elevated $pCO_2$ had no significant influence on DMSP production in *P. tricornuntum* and *T. weissflogii*."

L238: Not sure if "once-daily" is a proper English term

Thanks for the associate editor's suggestion. We have deleted this term in the revised manuscript.

L244-246 "This result has been observed in previous mesocosm experiments and it was attributed to small scale shifts in community composition and succession (Vogt et al., 2008; Webb et al., 2016)."

L239-249: Needs English copy-editing.

Thanks for the associate editor's suggestion. This part as well as the whole manuscript was edited by a native English speaker.

L246-256 "However, this phenomenon during the present study can be explained in another straightforward way. Previous studies have shown that marine bacteria play a key role in DMS production, and that the efficiency of bacteria converting DMSP to DMS may vary from 2 to 100% depending on the nutrient status of the bacteria and the quantity of dissolved organic matter (Simó et al., 2002, 2009; Kiene et al., 1999, 2000). In addition, a significant positive relationship was observed between DMS and DMSP-consuming bacteria (r = 0.643, $p$ < 0.01 in the low $p$CO$_2$ treatment; r = 0.544, $p$ < 0.01 in the high $p$CO$_2$ treatment) during this experiment. All of these observations point to the importance of bacteria in DMS and DMSP dynamics. During the present mesocosm experiment, DMSP concentrations in the low $p$CO$_2$ treatment decreased slightly on day 23, while the slight decrease appeared on day 29 in the high $p$CO$_2$ treatment (Fig. 2-B)."

L255: delete "the"

Thanks for the associate editor's suggestion. We have deleted "the" in the revised manuscript.

L262-265 "In addition, considering that algae and bacteria in natural seawater were removed through a filtering process before the experiment (Huang et al., 2018), we further concluded that the elevated $p$CO$_2$ controlled DMS concentrations mainly by affecting DMSP-consuming bacteria attached to *T. weissflogii* and *P. tricornuntum*."

L273: delete "have"

Thanks for the associate editor's suggestion. We have deleted "have" in the revised manuscript.

L278-279 "Previous studies reported that large-size cyanobacteria, such as *Aphanizomenon*

*flos-aquae*, produce bromocarbons (Karlsson et al., 2008)."

L274: Separate into two sentences: "...bromocarbons (Karlsson et al. 2008). Significant correlations..."

Thanks for the associate editor's suggestion. We have separated into two sentences in the revised manuscript.

L278-281 "Previous studies reported that large-size cyanobacteria, such as *Aphanizomenon flos-aquae*, produce bromocarbons (Karlsson et al., 2008). Significant correlations between the abundance of cyanobacteria and several bromocarbons have been reported in the Arabian Sea (Roy et al., 2011)."

L309-310: Change to: "...photochemical reaction could be responsible for the reduction of $CH_3I$ concentrations in the high $p$CO$_2$ treatment."

According to the associate editor's suggestion. We have modified this section in the revised manuscript.

"Both bacterial methyl transferase enzyme activity and photochemical reaction could be responsible for the reduction of $CH_3I$ concentrations in the high $p$CO$_2$ treatment but further experiments are needed to verify this result."

L315-316: Change to: "...was filtered prior to the experiment, algae from the coastal...were not present...".

According to the associate editor's suggestion mentioned above, we have deleted this sentence in the revised manuscript.

L318: Which phenomenon?

Thanks for the associate editor's suggestion. We have modified this section in the revised

manuscript.

L323-326 "In addition, a 4 d delay in DMS concentration was observed in the high $p$CO$_2$ treatment due to the effect of elevated $p$CO$_2$ and we attribute this delay in DMS concentration to the DMSP-consuming bacteria attached to *P. tricornuntum* and *T. weissflogii*."

L321-322: "as they were affected" is not proper English

Thanks for the associate editor's suggestion. We have modified this section in the revised manuscript.

"Affected by the filtration procedure, three bromocarbons compounds measured in this study were not correlated with *P. tricornuntum* and *T. weissflogii*, and Chl *a*."

L322: Start new sentence with "Elevated $p$CO$_2$..."

Thanks for the associate editor's suggestion. We have modified this section in the revised manuscript.

L327-328 "Besides, elevated $p$CO$_2$ had no effect on any of the three bromocarbons."

L478, 496, 503, 513: Replace "changes" by "development"

Thanks for the associate editor's suggestion. We have replaced "changes" by "development" in the revised manuscript.

"Fig. 1 Temporal development of pH in the high $p$CO$_2$ (1,000 µatm, solid squares) and low $p$CO$_2$ (400 µatm, white squares) mesocosms. Data are mean ± standard deviation, n = 3 (triplicate independent mesocosm bags) (Origin 8.0).

Fig. 2 Temporal development in dimethylsulfide (DMS), dimethylsulfoniopropionate (DMSP) and DMSP-consuming bacteria concentrations in the high $p$CO$_2$ (1,000 µatm, black squares) and low

$p$CO$_2$ (400 µatm, white squares) mesocosms. Data are mean ± standard deviation, n = 3 (triplicate

independent mesocosm bags).

Fig. 3 Temporal development in bromodichloromethane (CHBrCl$_2$), methyl bromide (CH$_3$Br),

dibromomethane (CH$_2$Br$_2$), iodomethane (CH$_3$I)   concentrations in the high $p$CO$_2$ (1,000 µatm,

black squares) and low $p$CO$_2$ (400 µatm, white squares) mesocosms. Data are mean ± standard

deviation, n = 3 (triplicate independent mesocosm bags)."

L517: Change to: "Dissolved inorganic carbon (DIC), ??? (pH$_T$), $p$CO$_2$ and nutrient concentrations

in the mesocosm experiments. "-" depicts that values were below detection limit." Define ???

**"Table 1**. Dissolved inorganic carbon (DIC), pH, $p$CO$_2$ and nutrient concentrations in the

mesocosm experiments. "-" means that the values were below the detection limit."

L520 and 528: What does "Relationship" mean? What are the numbers in the table? Ratios? Then

they should have no unit.

Thanks for the associate editor's suggestion. We have modified Table 2 and Table 3 in the revised

manuscript.

**Table 2**. Correlation between dimethylsulfide (DMS), dimethylsulfoniopropionate (DMSP), Chl $a$, bromodichloromethane (CHBrCl$_2$), methyl bromide (CH$_3$Br), dibromomethane (CH$_2$Br$_2$), iodomethane (CH$_3$I), DMSP-consuming bacteria, *Thalassiosira weissflogii* (*T. weissflogii*) and *Phaeodactylum tricornutum* (*P. tricornutum*) concentrations in the low $p$CO$_2$ treatments.

| | DMS | DMSP | Chl $a$ | CHBrCl$_2$ | CH$_3$Br | CH$_2$Br$_2$ | CH$_3$I | DMSP-consuming bacteria | *T. weissflogii* | *P. tricornutum* |
|---|---|---|---|---|---|---|---|---|---|---|
| DMS | 1 | | | | | | | | | |
| DMSP | 0.701** | 1 | | | | | | | | |
| Chl $a$ | 0.597** | 0.792** | 1 | | | | | | | |
| CHBrCl$_2$ | 0.526 | 0.280 | 0.559 | 1 | | | | | | |
| CH$_3$Br | -0.413 | -0.230 | 0.196 | 0.313 | 1 | | | | | |
| CH$_2$Br$_2$ | 0.310 | 0.180 | 0.001 | -0.136 | -0.308 | 1 | | | | |
| CH$_3$I | 0.694** | 0.654** | 0.717** | 0.596* | -0.151 | 0.129 | 1 | | | |
| DMSP-consuming bacteria | 0.643** | 0.520* | 0.522* | 0.394 | -0.268 | -0.038 | 0.762** | 1 | | |
| *T. weissflogii* | 0.410 | 0.617** | 0.899** | 0.301 | 0.322 | 0.028 | 0.680** | 0.399 | 1 | |
| *P. tricornutum* | 0.560* | 0.961** | 0.821** | 0.528 | -0.032 | 0.162 | 0.588** | 0.334 | 0.685** | 1 |

*. Correlation is significant at the 0.05 level (2-tailed).

**. Correlation is significant at the 0.01 level (2-tailed).

**Table 3**. Correlation between dimethylsulfide (DMS), dimethylsulfoniopropionate (DMSP), Chl $a$, bromodichloromethane (CHBrCl$_2$), methyl bromide (CH$_3$Br), dibromomethane (CH$_2$Br$_2$), iodomethane (CH$_3$I), DMSP-consuming bacteria, *Thalassiosira weissflogii* (*T. weissflogii*) and *Phaeodactylum tricornutum* (*P. tricornutum*) concentrations in the high $p$CO$_2$ treatments.

| | DMS | DMSP | Chl $a$ | CHBrCl$_2$ | CH$_3$Br | CH$_2$Br$_2$ | CH$_3$I | DMSP-consuming bacteria | *T. weissflogii* | *P. tricornutum* |
|---|---|---|---|---|---|---|---|---|---|---|
| DMS | 1 | | | | | | | | | |
| DMSP | 0.752[**] | 1 | | | | | | | | |
| Chl $a$ | 0.318[*] | 0.738[**] | 1 | | | | | | | |
| CHBrCl$_2$ | 0.324 | 0.094 | 0.326 | 1 | | | | | | |
| CH$_3$Br | -0.410 | -0.349 | 0.065 | 0.076 | 1 | | | | | |
| CH$_2$Br$_2$ | 0.540[*] | 0.352 | 0.142 | 0.233 | -0.377 | 1 | | | | |
| CH$_3$I | 0.694[**] | 0.816[**] | 0.741[**] | 0.690[*] | -0.407 | 0.316 | 1 | | | |
| DMSP-consuming bacteria | 0.544[*] | 0.522 | 0.549[*] | 0.532 | -0.311 | 0.368 | 0.851[*] | 1 | | |
| *T. weissflogii* | 0.355 | 0.743[**] | 0.930[**] | 0.304 | 0.076 | 0.233 | 0.690[**] | 0.567 | 1 | |
| *P. tricornutum* | 0.635[**] | 0.954[**] | 0.803[**] | 0.143 | -0.257 | 0.267 | 0.834[**] | 0.559 | 0.820[**] | 1 |

[*]. Correlation is significant at the 0.05 level (2-tailed).

[**]. Correlation is significant at the 0.01 level (2-tailed).

Fig. S1 caption: Would need to be changed to "J. Yu, unpubl. data". But will be unnecessary to cite since the data will be integrated into the main manuscript and hence become published data.

Thanks for the associate editor's suggestion. We have integrated this part into the main manuscript.

[Figure]

**Fig. 2** Temporal changes in dimethylsulfide (DMS), dimethylsulfoniopropionate (DMSP), DMSP-consuming

bacteria concentrations in the high $pCO_2$ (1,000 µatm, black squares) and low $pCO_2$ (400 µatm, white squares)

mesocosms. Data are mean ± standard deviation, n = 3 (triplicate independent mesocosm bags) (Origin 8.0).

**CERTIFICATE OF LANGUAGE EDITING**

The English writing of the following manuscript was carefully edited by a native English speaker.

**Manuscript Information**

| | |
|---|---|
| ID | AE201801070068-R2 |
| Editing date | 2018-09-30 |
| Title | Effect of elevated pCO2 on trace gas production during an ocean acidification mesocosm experiment |
| Corresponding author | Gui-peng Yang |
| Language writing before editing | □ Very poor    □ Poor    ■ Fair    □ Good    □ Very good    □ Excellent |
| Recommendation after language editing | □ Submitting to target journal directly
■ Submitting to target journal after minor revision
□ Re-editing required after major revision
□ Not suitable for publication |
| Overview comments | |

**Edited by**

**William K.**

Ph. D
North Carolina State University
Language Editing

**Certificate Issued by**

**Dr. Jason Qee**

Editor in Chief
Editorbar Language Editing, Beijing, China
runse@editorbar.com   www.editorbar.com

[Figure]

Certificate link: www.editorbar.com/order/cert/AE201801070068-R2

---

## Author Response (AR3)

**An itemized response (blue words) to the associate editor's comments and suggestions**

We are very thankful to the associate editor for the valuable comments and helpful suggestions to improve this manuscript. We have carefully considered the associate editor's suggestions and conducted the revision seriously.

Technical Corrections:

- Of Fig. 2 only A and B are cited. Please add a citation for 2-C.

Thanks for the associate editor's suggestion. We have added a citation for Fig. 2-C in the revised manuscript.

L253-254 "Similar to DMS, DMSP-consuming bacteria was also delayed in the high $pCO_2$ mesocosm compared to that in the low $pCO_2$ mesocosm (Fig. 2-C)."

- Of Fig. 3 only D is cited. Please add a citation for 3-A, 3-B, and 3-C

Thanks for the associate editor's suggestion. We have added a citation for Fig. 3-A, Fig. 3-B, and Fig. 3-C in the revised manuscript.

L262-263 "The temporal development in $CHBrCl_2$, $CH_3Br$, and $CH_2Br_2$ concentrations is shown in Fig. 3-A, Fig. 3-B, and Fig. 3-C, respectively."

- The sentence "Affected by the filtration procedure, three bromocarbons compounds measured in this study were not correlated with P. tricornuntum and T. weissflogii, and Chl a." (Line 322-323) is not clear to me. What is meant by "Affected by the filtration procedure"? Please provide a broader context.

The filtration procedure led to the loss of main bromocarbon production species, such as *Aphanizomenon flos-aquae*. In addition, the added *P. tricornuntum* and *T. weissflogii* did not primarily release these three bromocarbons during the mesocosm experiment. Therefore, three

bromocarbons compounds measured in this study were in low concentrations and not correlated with *P. tricornuntum* and *T. weissflogii*. We have explained this in the section 3.3, L276-284, and reworded this sentence in the revised manuscript.

L271-279 "No clear correlation was observed between the three bromocarbons and any of the measured algal groups (Table 2 and Table 3), indicating that *P. tricornuntum* and *T. weissflogii* did not primarily release these three bromocarbons during the mesocosm experiment. Previous studies reported that large-size cyanobacteria, such as *Aphanizomenon flos-aquae*, produce bromocarbons (Karlsson et al., 2008). Significant correlations between the abundance of cyanobacteria and several bromocarbons have been reported in the Arabian Sea (Roy et al., 2011). However, the filtration procedure led to the loss of cyanobacteria in the mesocosms and finally resulted in low bromocarbon concentrations during the experiment, although *P. tricornuntum* and *T. weissflogii* abundances were high."

L322-324 "Due to the loss of main bromocarbon-producing species affected by the filtration procedure, three bromocarbons compounds measured in this study were not correlated with *P. tricornuntum* and *T. weissflogii*, and Chl *a*."